# Hsp47 promotes biogenesis of multi-subunit neuroreceptors in the endoplasmic reticulum

Ya-Juan Wang[1†], Xiao-Jing Di[1†], Pei-Pei Zhang[1], Xi Chen[1], Marnie P Williams[1], Dong-Yun Han[1], Raad Nashmi[2], Brandon J Henderson[3], Fraser J Moss[1], Ting-Wei Mu[1*]

[1]Department of Physiology and Biophysics, Case Western Reserve University, Cleveland, United States; [2]Department of Biology, University of Victoria, Victoria, Canada; [3]Department of Biomedical Sciences, Marshall University, Huntington, United States

**Abstract** Protein homeostasis (proteostasis) deficiency is an important contributing factor to neurological and metabolic diseases. However, how the proteostasis network orchestrates the folding and assembly of multi-subunit membrane proteins is poorly understood. Previous proteomics studies identified Hsp47 (Gene: *SERPINH1*), a heat shock protein in the endoplasmic reticulum lumen, as the most enriched interacting chaperone for gamma-aminobutyric acid type A (GABA$_A$) receptors. Here, we show that Hsp47 enhances the functional surface expression of GABA$_A$ receptors in rat neurons and human HEK293T cells. Furthermore, molecular mechanism study demonstrates that Hsp47 acts after BiP (Gene: *HSPA5*) and preferentially binds the folded conformation of GABA$_A$ receptors without inducing the unfolded protein response in HEK293T cells. Therefore, Hsp47 promotes the subunit-subunit interaction, the receptor assembly process, and the antero-grade trafficking of GABA$_A$ receptors. Overexpressing Hsp47 is sufficient to correct the surface expression and function of epilepsy-associated GABA$_A$ receptor variants in HEK293T cells. Hsp47 also promotes the surface trafficking of other Cys-loop receptors, including nicotinic acetylcholine receptors and serotonin type 3 receptors in HEK293T cells. Therefore, in addition to its known function as a collagen chaperone, this work establishes that Hsp47 plays a critical and general role in the maturation of multi-subunit Cys-loop neuroreceptors.

**\*For correspondence:**
tingwei.mu@case.edu

[†]These authors contributed equally to this work

## Editor's evaluation

This important study defines new functions for the ER-resident protein HSP47 in the quality control of multi-pass membrane receptor proteins. The evidence supporting the conclusions is solid, with rigorous biochemical assays employed in appropriate models. However, additional consideration regarding the mechanism of HSP47-dependent regulation of membrane protein quality control would have strengthened the study. This work will be of broad interest to cell biologists and biochemists interested in the fields of proteostasis, membrane protein quality control, and neuroreceptor signaling.

## Introduction

Initially, the term 'molecular chaperone' was coined to describe a nuclear protein that enables the assembly of nucleosomes from folded histone proteins and DNA (*Ellis, 2013*; *Laskey et al., 1978*). Since then, the role of chaperones, including the heat shock proteins, in facilitating protein folding

(*Hartl et al., 2011*; *Horwich, 2014*) and maintaining protein homeostasis (proteostasis) at the cellular, tissue, and organismal levels has been extensively explored (*Balch et al., 2008*; *Balchin et al., 2016*; *Sala et al., 2017*). Proteostasis deficiencies have been recognized in a growing number of neurodegenerative, neurological, and metabolic diseases (*Ferro-Novick et al., 2021*; *Kelly, 2020*; *Wang and Kaufman, 2016*). Strategies to restore proteostasis, including applying regulators of the unfolded protein response (UPR) and Ca$^{2+}$ regulation, have been actively developed to ameliorate such protein conformational diseases (*Das et al., 2015*; *Grandjean and Wiseman, 2020*; *Mu et al., 2008*; *Tufanli et al., 2017*; *Wang et al., 2022a*). However, despite recent progress, the role of chaperones in regulating the folding and assembly of multi-subunit membrane proteins requires further elucidation (*Hegde, 2022*; *McKenna et al., 2020*).

Multi-subunit membrane protein assembly in the endoplasmic reticulum (ER) is intimately linked to their folding and ER-associated degradation (ERAD). The current limited knowledge about the assembly process was gained from studying various classes of membrane proteins, including dimeric T cell receptors (*Feige et al., 2015*), trimeric P2X receptors (*Boumechache et al., 2009*), trimeric sodium channels (*Buck et al., 2017*), tetrameric potassium channels (*Delaney et al., 2014*; *Li et al., 2017*), and pentameric nicotinic acetylcholine receptors (nAChRs) (*Green, 1999*; *Gu et al., 2016*). We use γ-aminobutyric acid type A (GABA$_A$) receptors as a physiologically important substrate to study their biogenesis (*Fu et al., 2016*). GABA$_A$ receptors are the primary inhibitory neurotransmitter-gated ion channels in mammalian central nervous systems (CNS; *Macdonald and Olsen, 1994*) and provide most of the inhibitory tone to balance the tendency of excitatory neural circuits to induce hyperexcitability, thus maintaining the excitatory-inhibitory balance (*Kirmse and Zhang, 2022*). Functional GABA$_A$ receptors are assembled as pentamers in the ER from eight subunit classes: α1–6, β1–3, γ1–3, δ, ε, θ, π, and $\rho$ 1–3. The most common subtype in the human brain contains two α1 subunits, two β2 subunits, and one γ2 subunit (*Sequeira et al., 2019*). To form a heteropentamer, individual subunits need to fold into their native structures in the ER (*Alder and Johnson, 2004*; *Skach, 2009*) and assemble with other subunits correctly on the ER membrane (*Barnes, 2001*; *Connolly et al., 1996a*; *Figure 1—figure supplement 1*). Only properly assembled pentameric receptors exit the ER, traffic through the Golgi for complex glycosylation, and reach the plasma membrane to perform their function. It was demonstrated that the α1 subunits fail to exit the ER on their own and are retained in the ER; after their assembly with β subunits, the α1β complex can exit the ER for subsequent trafficking to the plasma membrane (*Connolly et al., 1996a*; *Connolly et al., 1996b*). The inclusion of a γ2 subunit to form the pentamer further increases the conductance of the receptor and confers sensitivity to benzodiazepines (*Olsen and Sieghart, 2009*). Recently, it was reported that the synaptic localization of γ2-containing GABA$_A$ receptors requires the LHFPL family protein LHFPL4 and Neuroligin-2 (*Yamasaki et al., 2017*). However, many of the fundamental questions about how the proteostasis network regulates the multi-subunit membrane protein assembly process remains to be determined.

Elucidating the proteostasis network for the subunit folding and assembly process of multi-subunit membrane proteins and their biogenesis pathway in general is important to fine-tune their function in physiological and pathological conditions. Loss of function of GABA$_A$ receptors is one prominent cause of genetic epilepsies (*Hernandez and Macdonald, 2019*; *Hirose, 2014*). Furthermore, numerous variations in a single subunit cause subunit protein misfolding in the ER and/or disrupt assembly of the pentameric complex, leading to excessive ERAD, decrease cell surface localization of the receptor complex, and result in imbalanced neural circuits (*Fu et al., 2016*; *Fu et al., 2022*; *Wang et al., 2024*). The elucidation of the GABA$_A$ receptor proteostasis network will guide future efforts to develop strategies that restore proteostasis of variant GABA$_A$ receptors to ameliorate corresponding diseases, such as genetic epilepsies.

Recently, our quantitative affinity purification mass spectrometry-based proteomics analysis identified Hsp47 (Gene: *SERPINH1*) as the most enriched GABA$_A$ receptor-interacting chaperone (*Wang et al., 2022b*). Hsp47 is an ER-resident protein with a RDEL (Arg-Asp-Glu-Leu) ER retention signal (*Nagata et al., 1986*; *Saga et al., 1987*). Among the large Serpin (*se*rine *p*rotease *in*hibitor) superfamily, Hsp47 is the only one reported to show a molecular chaperone function (*Dafforn et al., 2001*). Current literature describes Hsp47 as a collagen-specific chaperone (*Nagata, 2003*; *Taguchi and Razzaque, 2007*). However, its broader role has been indicated (*Ito and Nagata, 2019*), such as interacting with the inositol-requiring enzyme 1α (IRE1α) to regulate the UPR (*Sepulveda et al., 2018*) as well as interacting with amyloid precursor protein (APP) in the CNS (*Bianchi et al., 2011*).

Here, we demonstrate that Hsp47 enhances the functional surface expression of endogenous GABA$_A$ receptors and other Cys-loop receptors in the CNS. Furthermore, a mechanistic study reveals that Hsp47 promotes the folding and assembly of multi-subunit neuroreceptors in the ER. Consequently, our results support a general role of Hsp47 in the protein quality control of multi-subunit Cys-loop receptors.

## Results

### Hsp47 directly interacts with GABA$_A$ receptor subunits

Since we previously identified Hsp47 as the most enriched GABA$_A$ receptor-interacting chaperone in HEK293T cells using quantitative proteomics (**Wang et al., 2022b**), here we evaluated the interaction between Hsp47 and GABA$_A$ receptors in more detail. Co-immunoprecipitation assays using mouse brain homogenates showed that the endogenous Hsp47 binds to endogenous GABA$_A$ receptor α1 subunits in the CNS (**Figure 1A**). Furthermore, to test a direct interaction between Hsp47 and GABA$_A$ receptor subunits, we carried out an in vitro binding assay using recombinant GST-tagged α1 or β2 subunits and recombinant His-tagged Hsp47. The anti-His antibody pulldown detected the α1 subunit in the GST-α1 complex (**Figure 1B**, lane 5) and the β2 subunit in the GST-β2 complex (**Figure 1B**, lane 10). No α1 or β2 bands were detected in the GST control complex (**Figure 1B**, lanes 4 and 9), indicating that Hsp47 directly binds to the GABA$_A$ receptor α1 and β2 subunits in vitro. Moreover, recombinant Hsp47 did not interact with recombinant hERG (human ether-a-go-go-related) potassium channels (**Vandenberg et al., 2012**), or recombinant ZIP7 (gene: *SLC39A7*), an ER membrane zinc efflux transporter (**Taylor et al., 2012**; **Figure 1—figure supplement 2A**), indicating that Hsp47 has certain selectivity to bind membrane proteins in vitro. These recombinant proteins were solubilized in detergents; however, caution needs to be taken since whether they adopt well-folded states is unclear.

Since Hsp47 resides in the ER lumen, it presumably interacts with the GABA$_A$ receptor ER luminal domain (ERD). The recombinant α1 subunit ERD and β2 subunit ERD adopted well-defined secondary structures according to circular dichroism experiments (**Figure 1—figure supplement 2B**), consistent with previous reports (**Shi et al., 2003**). Therefore, we determined the binding affinity between Hsp47 and α1(ERD) or β2(ERD). A MicroScale Thermophoresis (MST) assay reported strong interactions between Hsp47 and the ERD of GABA$_A$ receptor subunits: Kd (Hsp47-α1(ERD))=102 ± 10 nM; Kd (Hsp47-β2(ERD))=127 ± 15 nM (**Figure 1C**). Therefore, Hsp47 binds to GABA$_A$ receptor subunits with high affinity.

### Hsp47 positively regulates the functional surface expression of endogenous GABA$_A$ receptors in neurons

To the best of our knowledge, functional regulation of GABA$_A$ receptors and other ion channels by Hsp47 in the CNS has not been previously reported. Hsp47 is widely distributed in the CNS, including the cortex, hippocampus, hypothalamus, cerebellum, and olfactory bulb tested (**Figure 2—figure supplement 1A**)**,** which is consistent with the report that Hsp47 is robustly detected in primary cortical and hippocampal neurons and brain slices (**Bianchi et al., 2011**). Concomitantly, GABA$_A$ receptors are also distributed in these brain areas (**Figure 2—figure supplement 1A**; **Sequeira et al., 2019**; **Richards et al., 1987**).

Because GABA$_A$ receptors must reach the plasma membrane to act as ligand-gated ion channels, we first performed an indirect immunofluorescence microscopy experiment to evaluate how Hsp47 regulates their endogenous surface expression levels in primary rat hippocampal neurons. The application of anti-GABA$_A$ receptor subunit antibodies that recognize their extracellular N-termini without a prior membrane permeabilization step enabled us to label only the cell surface expressed proteins. Transduction of lentivirus carrying *SERPINH1* siRNA led to substantial depletion of Hsp47 in neurons (**Figure 2—figure supplement 1B**), and knocking down Hsp47 significantly decreased the surface staining of the major subunits of GABA$_A$ receptors, including the α1 subunits, β2/β3 subunits, and γ2 subunits (**Figure 2A**, row 2 to row 1). In addition, overexpressing Hsp47 by transduction of lentivirus carrying *SERPINH1* cDNA significantly enhanced the surface staining of endogenous α1, β2/β3, and γ2 subunits in neurons (**Figure 2B**, row 2 to row 1). These results indicated that Hsp47 positively regulates the surface protein levels of endogenous GABA$_A$ receptors. Furthermore, whole-cell patch-clamp

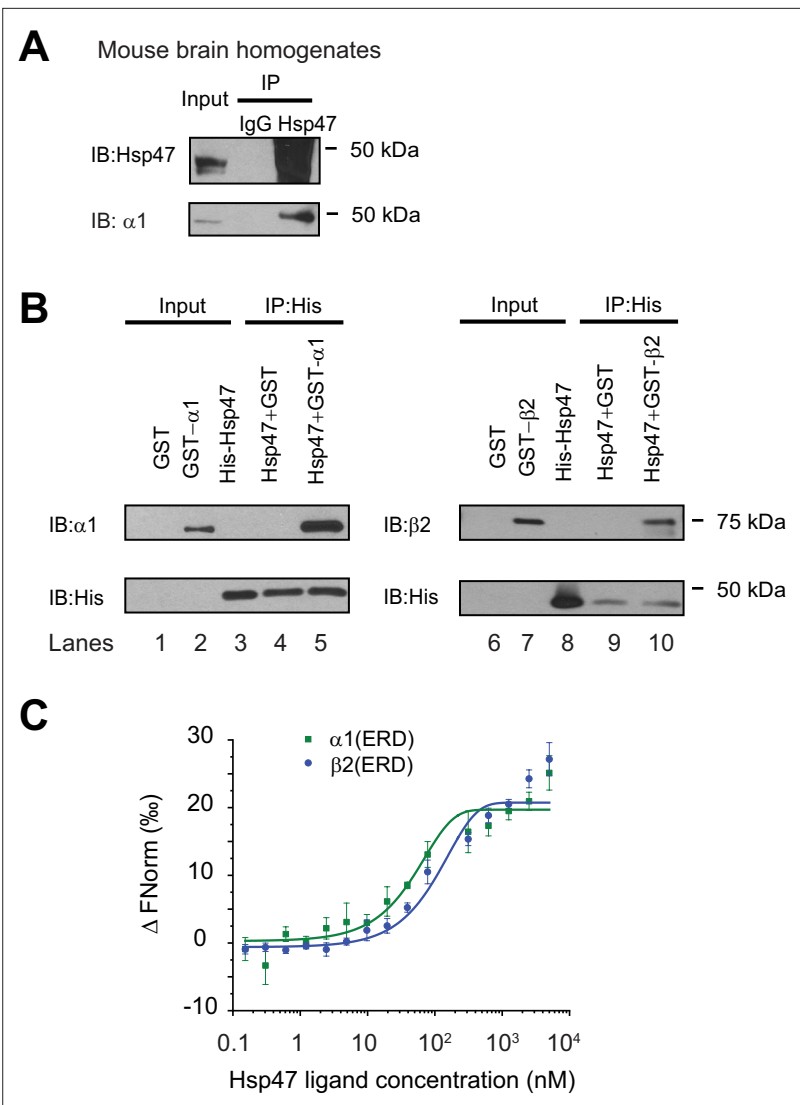

**Figure 1.** Hsp47 interacts with GABA_A receptors. (**A**) Endogenous interactions between GABA_A receptor α1 subunits and Hsp47. Mouse brain homogenates from 8 to 10 weeks C57BL/6 J mice were immunoprecipitated with an anti-α1 antibody, and the immunoisolated eluents were blotted with indicated antibodies. IgG was included as a negative control for non-specific binding. Three biological replicates were performed. (**B**) Recombinant Hsp47 binds recombinant α1 subunit and β2 subunit of GABA_A receptors in vitro. GST, GST-tagged α1 or GST-tagged β2 recombinant protein was mixed with His-tagged Hsp47 in buffers containing 1% Triton X-100. The protein complex was isolated by immunoprecipitation using an anti-His antibody, and the immunopurified eluents were separated by SDS-PAGE and blotted with indicated antibodies. Three biological replicates were performed. (**C**) MicroScale Thermophoresis (MST) was used to determine the binding affinities between Hsp47, an ER luminal chaperone, to RED-labeled His-α1(ERD) and His-β2(ERD). Increasing concentrations of recombinant Hsp47 proteins (0.2 nM – 10 µM) were incubated with 50 nM RED-labeled His-α1(ERD) or His-β2(ERD) in PBS with Tween-20 (0.05%). Then samples were loaded to the capillaries and measured using a Monolith NT.115 instrument with the settings of 40% LED/excitation and 40% MST power. Three biological replicates were performed. The data were analyzed using the Monolith software for the calculation of the dissociation constant (Kd). IP, immunoprecipitation; IB, immunoblotting.

The online version of this article includes the following source data and figure supplement(s) for figure 1:

**Source data 1.** Original files for the western blot analysis in *Figure 1A and B*.

**Source data 2.** PDF containing the original blots in *Figure 1A and B* with the relevant bands clearly labeled.

**Figure supplement 1.** The GABA_A receptor biogenesis pathway.

**Figure supplement 2.** In vitro interactions between Hsp47 and membrane proteins.

*Figure 1 continued on next page*

*Figure 1 continued*

**Figure supplement 2—source data 1.** Original files for the western blot analysis in *Figure 1—figure supplement 2A*.

**Figure supplement 2—source data 2.** PDF containing the original blots in *Figure 1—figure supplement 2A* with the relevant bands clearly labeled.

electrophysiology recordings demonstrated that depleting Hsp47 significantly decreased the peak GABA-induced currents from 1660 ± 413 pA in the presence of scrambled siRNA to 886±157 pA after the application of lentivirus carrying *SERPINH1* siRNA in hippocampal neurons, whereas over-expressing Hsp47 increased the peak current amplitude to 2455±406 pA in hippocampal neurons (*Figure 2C*). Collectively, the experiments in *Figure 2* unambiguously reveal a novel role of Hsp47 as a positive regulator of the functional surface expression of endogenous GABA$_A$ receptors, an important neuroreceptor.

## Hsp47 preferentially binds the folded conformation of GABA$_A$ receptor subunits and promotes their ER-to-Golgi trafficking

Since Hsp47 is an ER luminal chaperone, we hypothesized that to enhance the surface trafficking of GABA$_A$ receptors, Hsp47 promotes their protein folding in the ER and subsequent anterograde traf-ficking. We used an endoglycosidase H (Endo H) enzyme digestion assay to monitor the ER-to-Golgi trafficking of GABA$_A$ receptors, also as a surrogate to determine whether GABA$_A$ receptors are folded and assembled properly in the ER (*Di et al., 2013*). Due to the heterogeneity of GABA$_A$ receptor subunits in neurons, we employed HEK293T cells to exogenously express the major subtype of GABA$_A$ receptors containing α1, β2, and γ2 subunits for the mechanistic study (*Taylor et al., 2000*). The Endo H enzyme selectively cleaves after asparaginyl-*N*-acetyl-D-glucosamine (GlcNAc) in the N-linked glycans in the ER, but it cannot remove this oligosaccharide chain after the high mannose form is enzymatically remodeled in the Golgi. Therefore, Endo H resistant subunit bands represent properly folded and assembled, post-ER subunit glycoforms, which traffic at least to the Golgi. Since the α1 subunit has two N-glycosylation sites at Asn38 and Asn138, Endo H digestion generated two Endo H-resistant α1 bands (*Figure 3A*, lanes 2 and 4, top two bands), corresponding to singly and doubly glycosylated α1 proteins, which were observed in previous experiments (*Han et al., 2015b*). The Endo H digestion assay showed that Hsp47 overexpression increased the Endo H resistant band intensity (*Figure 3A*, lane 4 to lane 2) and the ER to Golgi traffic king efficiency, represented by the ratio of the Endo H resistant α1 band / total α1 band (*Figure 3A*, lane 4 to lane 2; quantification shown in the *Figure 3A* bottom panel). This result indicated that Hsp47 enhanced the folding and assembly of GABA$_A$ receptors in the ER and thus their ER-to-Golgi trafficking. Consistently, Hsp47 overexpression increased the peak current 1.6-fold in HEK293T cells expressing α1β2γ2 receptors (*Figure 3—figure supplement 1A*). In addition, cellular ubiquitination assay demonstrated that Hsp47 overexpression decreased ubiquitinated α1 protein level (*Figure 3—figure supplement 1B*), suggesting that Hsp47 reduced the population of misfolded α1 proteins. Cycloheximide-chase assay showed that over-expressing Hsp47 (*Figure 3—figure supplement 1C*) or knocking down Hsp47 (*Figure 3—figure supplement 1D*) did not change the apparent degradation rate of α1 proteins significantly. More-over, co-immunoprecipitation assay showed that knocking down Hsp47 did not significantly influence the interactions between α1 and BiP, an Hsp70 family chaperone in the ER lumen (*Figure 3—figure supplement 1E*), suggesting the involvement of additional ER proteostasis network components in handling misfolded α1 proteins.

We next evaluated how Hsp47 coordinates the folding and assembly of GABA$_A$ receptors. We perturbed α1 subunit folding both genetically and chemically in HEK293T cells expressing GABA$_A$ receptors. We then evaluated the correlation between the relative folding degree of the α1 subunits and their interaction with Hsp47 using a co-immunoprecipitation assay. Individual GABA$_A$ receptor subunits have a signature disulfide bond in their large N-terminal domain. Adding dithiothreitol (DTT), which is cell-permeable, to the cell culture media for 10 min destroyed the signature disulfide bond between Cys166 and Cys180 in the α1 subunit, thus compromising its folding. This operation did not change the total α1 protein levels (*Figure 3B*, lanes 2–4 to lane 1) possibly because, during such a short time, degradation of the misfolded α1 subunit was not substantial. In sharp contrast, adding DTT significantly decreased the α1 protein that was pulled down by Hsp47 in a dose-dependent

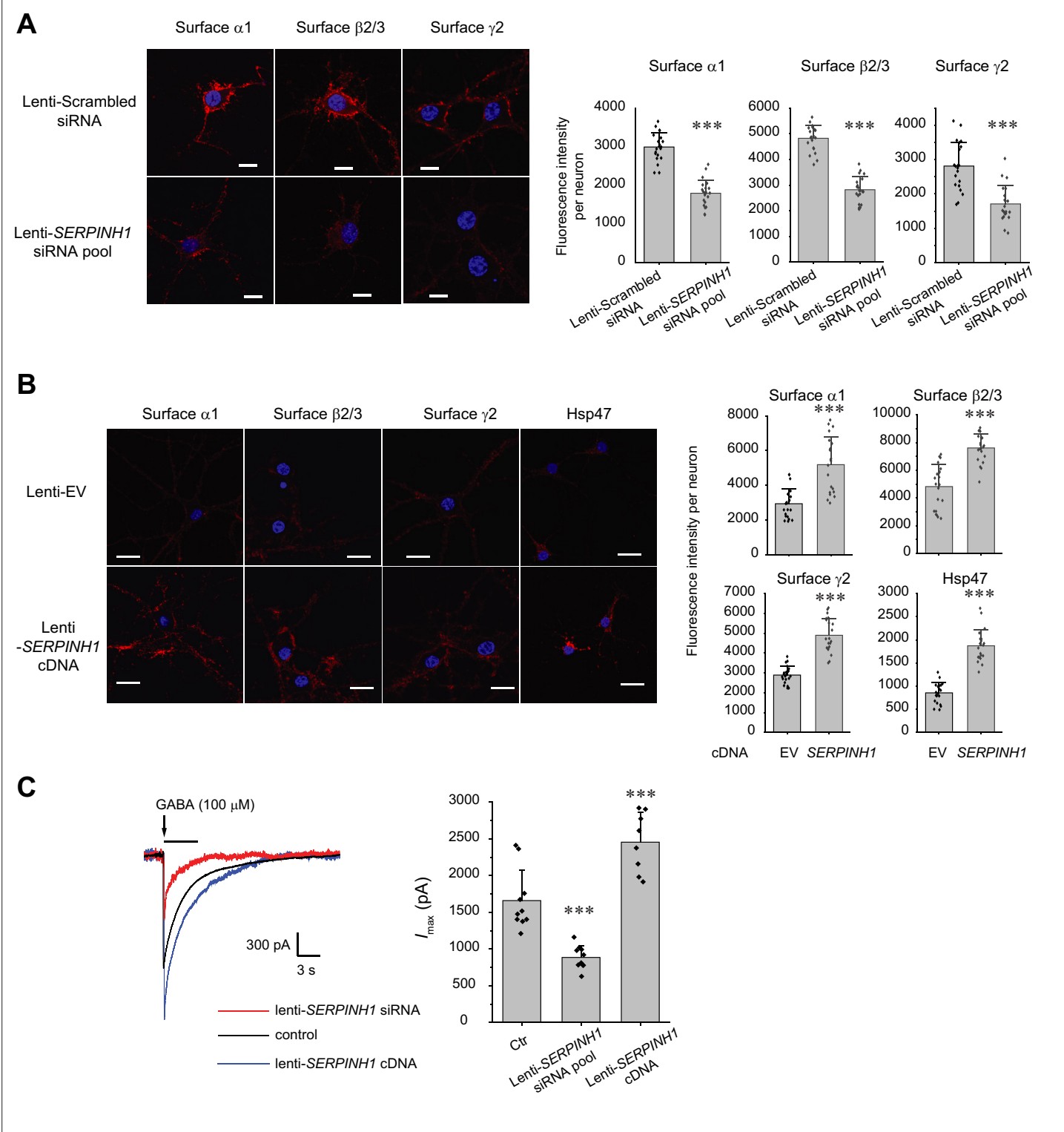

**Figure 2.** Hsp47 positively regulates the surface expression of endogenous GABA$_A$ receptors in cultured neurons. (**A, B**) Effect of knocking down Hsp47 (**A**) and overexpressing Hsp47 (**B**) on the surface expression of endogenous GABA$_A$ receptor subunits in primary rat hippocampal neurons. Cultured neurons were transduced with *SERPINH1* siRNA lentivirus or scrambled siRNA lentivirus (**A**) and with *SERPINH1* cDNA lentivirus or empty vector (EV) lentivirus (**B**) at days in vitro (DIV) 10. Forty-eight hours post transduction, surface GABA$_A$ receptors were stained using anti-α1 subunit, anti-β2/β3 subunit, or anti-γ2 subunit antibodies without membrane permeabilization. The cells were then washed, and permeabilized before we stained the nuclei with DAPI. Hsp47 staining was carried out after membrane permeabilization. At least 20 neurons from at least three transductions were imaged

*Figure 2 continued on next page*

*Figure 2 continued*

by confocal microscopy for each condition. Representative images are shown on the left side. Scale bar = 10 µm (**A**) or 20 µm (**B**). Quantification of the fluorescence intensity of the surface GABA$_A$ receptor subunits or Hsp47 after background correction per neuron was shown on the right. (**C**) Whole-cell patch clamping was performed to record GABA-induced currents. Neurons were subjected to transduction as in (**A**) and (**B**). The recordings were carried out 48 hr post transduction. Eight to ten neurons from three transductions were recorded. Representative traces are shown in the left-hand panel. Peak current amplitude ($I_{max}$) is shown on the right. The holding potential was set at −60 mV. pA: picoampere. Each data point is reported as mean ± SD. Statistical significance was calculated using t-test (**A, B**) or one-way ANOVA followed by post hoc Tukey's HSD test (**C**). *** p<0.001.

The online version of this article includes the following source data and figure supplement(s) for figure 2:

**Source data 1.** Data used for graphs presented in *Figure 2A, B and C*.

**Figure supplement 1.** Hsp47 expression in the central nervous system.

**Figure supplement 1—source data 1.** Original files for the western blot analysis in *Figure 2—figure supplement 1A*.

**Figure supplement 1—source data 2.** PDF containing the original blots in *Figure 2—figure supplement 1A* with the relevant bands clearly labeled.

**Figure supplement 1—source data 3.** Data used for graphs presented in *Figure 2—figure supplement 1B*.

manner (*Figure 3B*, lanes 7–9 to lane 6, quantification shown in the bottom panel). This indicates that eliminating the signature disulfide bond in the α1 subunit decreased its interaction with Hsp47 and supports the hypothesis that Hsp47 preferentially binds to the folded α1 subunit conformation.

In addition, we genetically disrupted the signature disulfide bond in the α1 subunit either by introducing a single C166A mutation or C166A/C180A double mutations. The co-immunoprecipitation assay clearly demonstrated that both the single and double mutations led to a decreased interaction between the α1 subunit and Hsp47 (*Figure 3C*, lanes 6 and 7 to lane 5, quantification shown in the bottom panel). To evaluate the relative conformational stability of the α1 subunit variants, we examined the Triton X-100 detergent soluble fractions and the Triton X-100 detergent-insoluble fractions. The percentage of the insoluble fractions in the C166A single mutation and the C166A/C180A double mutations is significantly greater than that in the WT receptors (*Figure 3D*, quantification shown in the bottom panel), indicating that disrupting the signature disulfide bond induces aggregation. Notably, the C166A single mutant is more prone to aggregation than the C166A/C180A double mutant since the single C166A mutant subunit retains an unpaired Cys180 in the ER lumen that remains available for cross-linking.

During the biogenesis in the ER, GABA$_A$ receptors need to interact with a network of chaperones and folding enzymes, such as BiP, to acquire their native structures (*Wang et al., 2022b*). BiP, which binds the hydrophobic patches of unfolded proteins and prevents their aggregation (*Flynn et al., 1991*; *Otero et al., 2010*), interacts with the GABA$_A$ receptors in the ER (*Connolly et al., 1996a*; *Di et al., 2013*). We reasoned that if Hsp47 preferentially binds the folded conformation of the GABA$_A$ receptor subunits, it would act after BiP because BiP is expected to act early in the protein folding step in the ER (*Knittler and Haas, 1992*; *Melnick et al., 1994*). We therefore evaluated how disrupting appropriate α1 subunit folding influenced their interaction with BiP. As hypothesized, the interactions between the α1 subunits and BiP were significantly enhanced when the signature disulfide bonds were chemically destroyed by adding DTT to the cell culture media (*Figure 3E*, lanes 7, 8, and 9 to lane 6, quantification shown in the bottom panel). Genetic destruction of the α1 subunit disulfide bonds in the C166A single mutant or the C166A/C180A double mutant (*Figure 3F* lanes 6 and 7 to lane 5, quantification shown in the bottom panel) produced similar results. Collectively, *Figure 3* indicates that inducing misfolding of GABA$_A$ receptors compromises their interactions with Hsp47, whereas, in sharp contrast, enhances their interactions with BiP. Therefore, BiP preferentially binds the unfolded/misfolded states, whereas Hsp47 preferentially binds the properly folded states of the α1 subunits. Hsp47 must therefore acts after BiP to enhance the productive folding of GABA$_A$ receptors.

## Hsp47 enhances the subunit-subunit assembly of GABA$_A$ receptors

A cellular environment is required for the assembly of the majority of ion channels (*Green, 1999*), indicating that factors other than ion channel subunits themselves are necessary in this process. We next tested our hypothesis that Hsp47 promotes efficient GABA$_A$ receptor subunit assembly. We used Förster resonance energy transfer (FRET) to evaluate the cellular interactions between GABA$_A$ receptor subunits. We incorporated enhanced cyan fluorescent protein (CFP) (donor) into the TM3-TM4 intracellular loop of the α1 subunit and enhanced yellow fluorescent protein (YFP) (acceptor) into

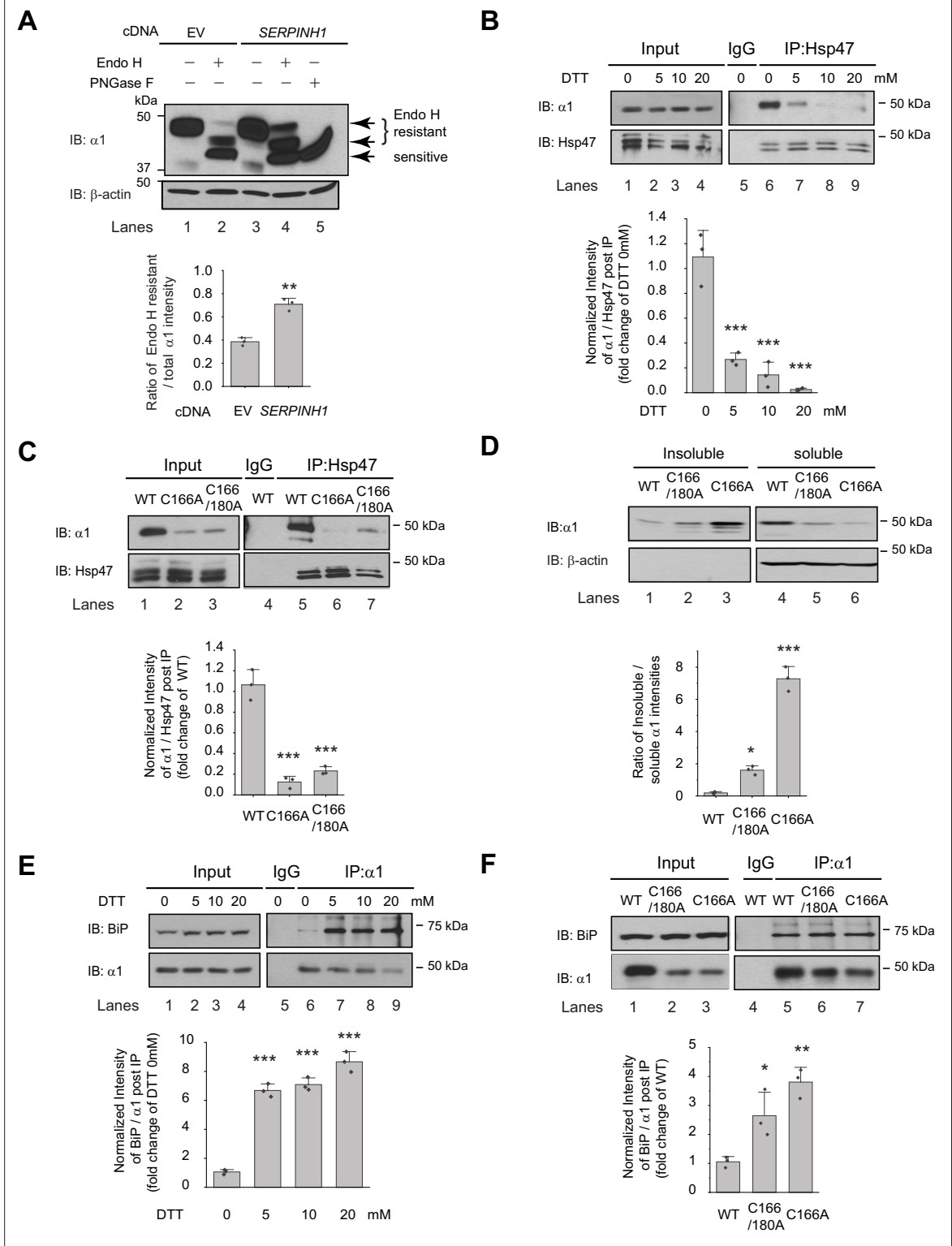

**Figure 3.** Hsp47 preferentially binds the folded conformation of GABA_A receptor subunits. (**A**) Overexpression of Hsp47 increases the endo H-resistant post-ER glycoform of the α1 subunit in HEK293T cells stably expressing α1β2γ2 GABA_A receptors. The peptide-N-glycosidase F (PNGase F) enzyme cleaves the innermost GlcNAc and serves a control for unglycosylated α1 proteins (lane 5). Two endo H-resistant bands were detected for the α1 subunit since there are two N-glycosylation sites in α1, indicated by the bracket (lanes 2 and 4). Quantification of the ratio of endo H-resistant / total α1 subunit bands, as a measure of the ER-to-Golgi trafficking efficiency, is shown on the bottom. (**B**) Dithiothreitol (DTT) treatment decreases the interaction between Hsp47 and α1 subunit of GABA_A receptors. HEK293T cells stably expressing WT α1β2γ2 GABA_A receptors were treated with

*Figure 3 continued on next page*

*Figure 3 continued*

indicated concentration of DTT in the PBS buffer for 10 min. Then Triton X-100 cell extracts were immunoprecipitated with a mouse anti-Hsp47 antibody, and the immunoisolated eluents were subjected for immunoblotting assay. Quantification of the relative intensity of α1/Hsp47 post IP, as a measure of their interactions, is shown on the bottom panel. (**C**) Disulfide bond mutations in the α1 subunit decrease the interaction between Hsp47 and α1 subunit of GABA$_A$ receptors. HEK293T cells were transiently transfected with WT α1β2γ2, α1(C166A)β2γ2, or α1(C166A, C180A)β2γ2 subunits. Forty-eight hours post transfection, Triton X-100 cell extracts were immunoprecipitated with a mouse anti-Hsp47 antibody, and the immunoisolated eluents were subjected for immunoblotting assay. Quantification of the relative intensity of α1/Hsp47 post IP is shown on the bottom panel. (**D**) Disulfide bond mutations in the α1 subunits decrease the solubility of the α1 subunit protein. HEK293T cells were transiently transfected as in (**C**). Forty-eight hours post transfection, the Triton X-100 detergent soluble fractions and the Triton X-100 detergent insoluble fractions were isolated for immunoblotting assay. Quantification of the ratio of insoluble/soluble fractions, as a measure of relative aggregation, is shown on the bottom panel. (**E**) DTT treatment increases the interaction between BiP and α1 subunit of GABA$_A$ receptors. HEK293T cells stably expressing α1β2γ2 GABA$_A$ receptors were treated with indicated concentrations of DTT in PBS for 10 minutes. Then Triton X-100 cell extracts were immunoprecipitated with a mouse anti-α1 antibody, and the immunoisolated eluents were subjected for immunoblotting assay. Quantification of the relative intensity of BiP/α1 post IP is shown on the bottom panel. (**F**) The disulfide mutations of α1 subunit increase the interaction between BiP and the α1 subunit. HEK293T cells were transiently transfected as in (**C**). Forty-eight hours post transfection, Triton X-100 cell extracts were immunoprecipitated with a mouse anti-α1 antibody, and the immunoisolated eluents were subjected for immunoblotting assay. Quantification of the relative intensity of BiP/α1 post IP is shown on the bottom panel. IP, immunoprecipitation; IB, immunoblotting. For (**A**)-(**F**), three biological replicates were performed. Each data point is reported as mean ± SD. Significant difference was analyzed by t-test (**A**), or a one-way ANOVA followed by post hoc Tukey's HSD test (**B–F**). *, $p<0.05$; **, $p<0.01$; ***, $p<0.001$.

The online version of this article includes the following source data and figure supplement(s) for figure 3:

**Source data 1.** Original files for the western blot analysis in *Figure 3A, B, C, D, E and F*.

**Source data 2.** PDF containing the original blots in *Figure 3* with the relevant bands clearly labeled.

**Source data 3.** Data used for graphs presented in *Figure 3A, B, C, D, E and F*.

**Figure supplement 1.** Effect of Hsp47 on the degradation of wild type GABA$_A$ receptors.

**Figure supplement 1—source data 1.** Original files for the western blot analysis in *Figure 3—figure supplement 1B*, C, D, and E.

**Figure supplement 1—source data 2.** PDF containing the original blots in *Figure 3—figure supplement 1* with the relevant bands clearly labeled.

**Figure supplement 1—source data 3.** Data used for graphs presented in *Figure 3—figure supplement 1A*, B, and E.

the TM3-TM4 intracellular loop of the β2 subunit. The addition of CFP/YFP into the large intracellular loops of GABA$_A$ receptors did not change the function of GABA$_A$ receptors since dose response to GABA was indistinguishable between α1β2γ2 receptors and (CFP-α1)(YFP-β2)γ2 receptors according to patch-clamp electrophysiology recordings in HEK293T cells (*Figure 4—figure supplement 1*). The TM3-TM4 intracellular loops are the most variable segment within GABA$_A$ receptor subunits and their splice variants. These were often replaced with short sequences in structural studies (*Laverty et al., 2019*; *Zhu et al., 2018*). Intracellular loops that incorporated CFP or YFP were also utilized in FRET experiments performed on nAChRs (*Nashmi et al., 2003*), members of the same Cys-loop superfamily to which GABA$_A$ receptors belong. Pixel-based FRET experiments showed that mean FRET efficiency was 24.4 ± 3.1% for (CFP-α1)(YFP-β2)γ2 receptors (*Figure 4A*, row 1, column 3); overexpressing Hsp47 significantly increased the mean FRET efficiency to 31.7 ± 5.7% (*Figure 4A*, row 2, column 3), indicating that Hsp47 positively regulates the assembly between α1 and β2 subunits of GABA$_A$ receptors. In addition, the co-immunoprecipitation assay showed that overexpression of Hsp47 significantly increased the relative amount of the β2 subunit that was pulled down with the α1 subunit (*Figure 4B*, lane 5–4, quantification shown on the bottom), indicating that Hsp47 promotes incorporation of β2 subunits into GABA$_A$ receptor pentamers.

Further, we used non-reducing protein gels to evaluate how Hsp47 influences the formation of the oligomeric subunits during the assembly process in the ER. The absence of reducing reagents in the protein gel's sample loading buffer preserves the intra- and inter-subunit disulfide bonds, which is expected to enable the detection of subunit oligomerization. In HEK293T cells expressing α1β2 receptors, distinct bands around 480 kDa were visible for both α1 subunit and β2 subunit in non-reducing gels (*Figure 4C*, lanes 2–5), indicating that the 480 kDa complex corresponds to the α1β2 hetero-oligomers. Moreover, the apparent molecular weight of the 480 kDa complex agrees with the molecular weight of the detected native GABA$_A$ receptors obtained from the cerebellum using blue native protein gels (*Yamasaki et al., 2017*). Therefore, the 480 kDa complex probably corresponds to the correctly assembled receptor complex. Strikingly, overexpression of Hsp47 increased the intensity of the 480 kDa bands for both α1 and β2 subunits (*Figure 4C*, lanes 3–5 to lane 2, quantification in *Figure 4D*), indicating that Hsp47 promotes the formation of the properly assembled oligomers. In

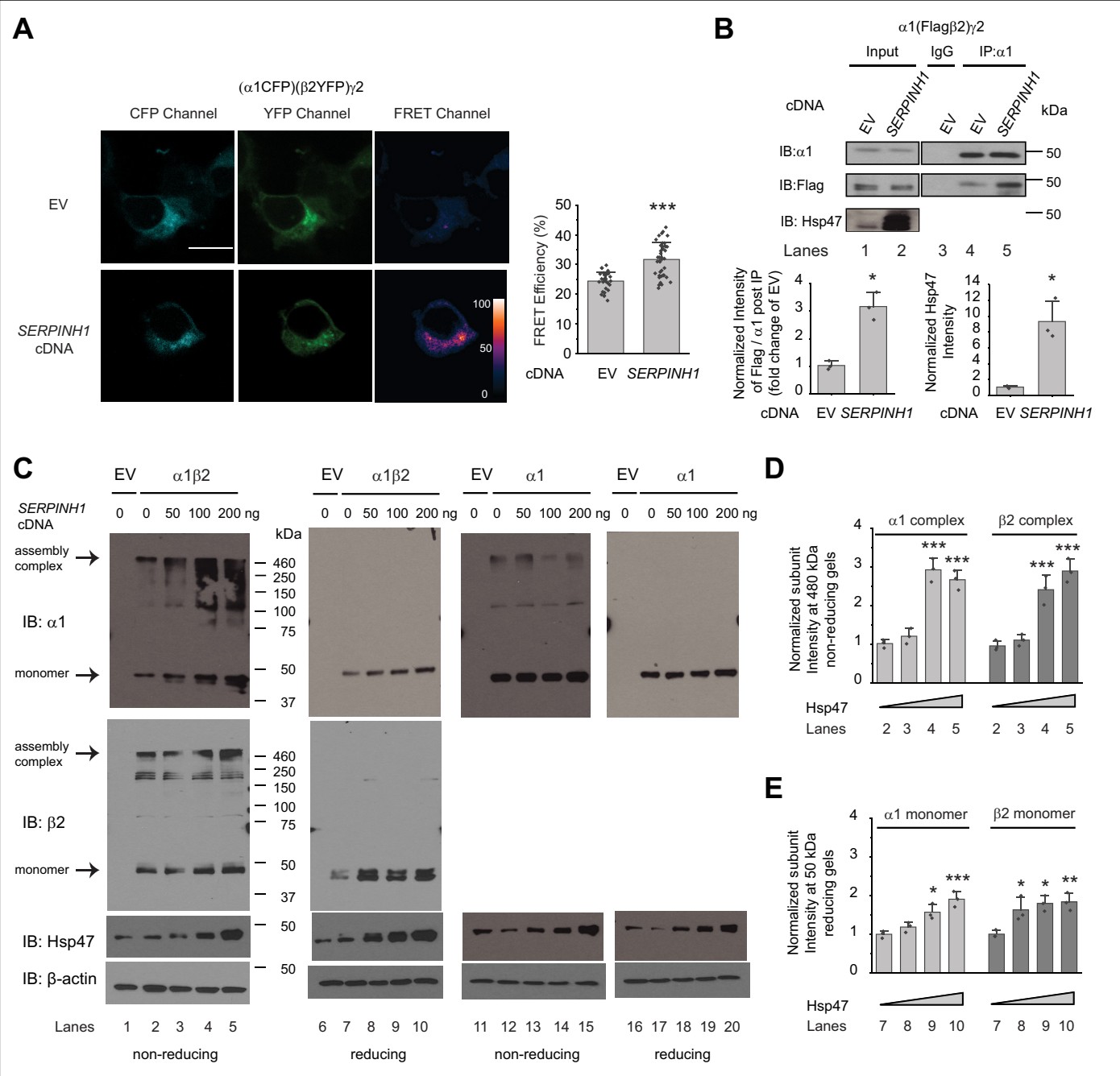

**Figure 4.** Hsp47 promotes the assembly of GABA$_A$ receptors. (**A**) Hsp47 overexpression increases FRET efficiency between CFP-tagged α1 subunit and YFP-tagged β2 subunit of GABA$_A$ receptors. HEK293T cells were transfected with CFP-tagged α1 subunit, YFP-tagged β2 subunit, and γ2 subunit; in addition, cells were transfected with empty vector (EV) control or Hsp47 cDNA. Forty-eight hours post transfection, pixel-based FRET was used to measure the FRET efficiency between α1-CFP and β2-YFP by using a confocal microscope. Representative images were shown for the CFP channel (1st columns), YFP channel (2nd columns), and FRET efficiency (3rd columns). Scale bar = 10 μm. Quantification of the FRET efficiency from 30 to 41 cells from at least three transfections was achieved using the ImageJ PixFRET plug-in, and shown on the right. (**B**) Overexpression of Hsp47 increases the interaction between α1 and β2 subunit of GABA$_A$ receptors. HEK293T cells stably expressing α1(Flag-β2)γ2 GABA$_A$ receptors were transfected with empty vector (EV) control or *SERPINH1* cDNA. Forty-eight hours post transfection, Triton X-100 cell extracts were immunoprecipitated with a mouse anti-α1 antibody, and the immunoisolated eluents were subjected to immunoblotting assay. Three biological replicates were performed. Quantification of the relative intensity of Flag-β2 / α1 post IP is shown on the bottom. (**C**) HEK293T cells were transiently transfected with empty vector (EV), α1 subunits alone, or both α1 and β2 subunits of GABA$_A$ receptors together with *SERPINH1* cDNA plasmids at various concentrations. Forty-eight hours post transfection, cells were lysed in RIPA buffer, and the total cell lysates were subjected to SDS-PAGE under non-reducing conditions and reducing conditions and immunoblotting analysis. Three biological replicates were performed. (**D**) Quantification of the 480 kDa band intensities for α1 and β2

*Figure 4 continued on next page*

*Figure 4 continued*

subunits under non-reducing conditions (lanes 2–5 in **C**) (n=3). (**E**) Quantification of the 50 kDa band intensities for α1 and β2 subunits under reducing conditions (lanes 7–10 in **C**) (n=3). IP, immunoprecipitation; IB, immunoblotting. Each data point is reported as mean ± SD. Significant difference was analyzed by t-test (**A, B**) or a one-way ANOVA followed by post hoc Tukey's HSD test (**D, E**). *, p<0.05; **, p<0.01; ***, p<0.001.

The online version of this article includes the following source data and figure supplement(s) for figure 4:

**Source data 1.** Original files for the western blot analysis in *Figure 4B and C*.

**Source data 2.** PDF containing the original blots in *Figure 4* with the relevant bands clearly labeled.

**Source data 3.** Data used for graphs presented in *Figure 4A, B, D and E*.

**Figure supplement 1.** Dose-response curves of GABA$_A$ receptors.

addition, reducing protein gels showed that overexpressing Hsp47 increased the band intensities for α1 and β2 subunits at 50 kDa in HEK293T cells expressing α1β2 receptors (*Figure 4C*, lanes 8–10 to lane 7, quantification in *Figure 4E*). Moreover, we carried out control experiments using HEK293T cells expressing only α1 subunits because α1 subunits alone cannot exit the ER (*Connolly et al., 1996b*). Non-reducing gels revealed that the majority of the detected α1 protein was in the monomeric form (*Figure 4C*, lane 12), and overexpression of Hsp47 did not change the intensity of α1 subunit band on the non-reducing gels (*Figure 4C*, lanes 13–15 to lane 12) or using reducing gels (*Figure 4C*, lanes 18–20 to lane 17). This probably occurred because α1 subunits alone cannot assemble to form a trafficking-competent complex to exit the ER. Collectively, these results indicated that Hsp47 promotes the assembly of the native pentameric GABA$_A$ receptor complexes for their subsequent ER exit and trafficking to the Golgi and plasma membrane.

## Overexpressing Hsp47 rescues functional surface expressions of trafficking-deficient, epilepsy-associated GABA$_A$ receptor variants

We next evaluated the effect of Hsp47 on the function of pathogenic GABA$_A$ receptors harboring trafficking-deficient variations in the α1 subunit. A well-characterized misfolding-prone α1(A322D) variant possesses an extra negative charge in the TM3 helix of the α1 subunit, leading to the inefficient insertion of TM3 into the lipid bilayer and its fast degradation (*Di et al., 2013*; *Gallagher et al., 2007*). As a result, the α1(A322D) variant causes loss of function of GABA$_A$ receptors and juvenile myoclonic epilepsy (*Cossette et al., 2002*). An Endo H enzyme digestion assay showed that overexpressing Hsp47 increased the ratio of Endo H resistant α1/total α1 bands from 0.20±0.03–0.35±0.08 (*Figure 5A*, lane 4–2), indicating that Hsp47 promoted the formation of properly folded and assembled GABA$_A$ receptors in the ER and increased the trafficking efficiency of the α1(A322D) variant from the ER to the Golgi. Consistently, Hsp47 overexpression substantially reduced the heavily ubiquitinated α1(A322D) protein (*Figure 5B*), indicating that Hsp47 decreased the population of misfolded α1(A322D) protein. Cycloheximide-chase assay demonstrated that overexpressing Hsp47 (*Figure 5—figure supplement 1A*) or knocking down Hsp47 (*Figure 5—figure supplement 1B*) did not change the apparent degradation rate of α1(A322D) significantly. In addition, co-immunoprecipitation assay showed that knocking down Hsp47 did not increase the interactions between BiP and α1(A322D) (*Figure 5—figure supplement 1C*), suggesting the involvement of additional proteostasis network components in handling misfolded α1(A322D).

Furthermore, overexpressing Hsp47 significantly increased α1(A322D) variant surface expression according to surface biotinylation assay and the total α1(A322D) protein levels (*Figure 5C*). The Hsp47-enhanced α1(A322D) surface expression was also reflected in the GABA-induced $I_{max}$ from 8.5±4.4 pA (n=20) to 50.3±19.7 pA (n=17) in HEK293T cells expressing α1(A322D)β2γ2 GABA$_A$ receptors (*Figure 5D*). Therefore, Hsp47 promotes the functional surface expression of an epilepsy-associated GABA$_A$ receptor variant.

Moreover, we evaluated the effect of Hsp47 on additional trafficking-deficient α1 variants, including α1(S76R), α1(D219N), and α1(G251D) (*Figure 5E*; *Fu et al., 2022*). Previously, it was reported that these variations decreased the surface expression of the α1 subunits and reduced GABA-induced peak current amplitudes to 33.3% for α1(S76R), to 60.3% for α1(D219N), and to 49.2% for α1(G251D) compared to wild-type receptors (*Wang et al., 2023*). Overexpressing Hsp47 (*Figure 5—figure supplement 2*) significantly promoted the surface expression of these α1 variants in HEK293T cells (*Figure 5F*). Furthermore, Hsp47 overexpression increased the peak currents 1.59-fold in HEK293T

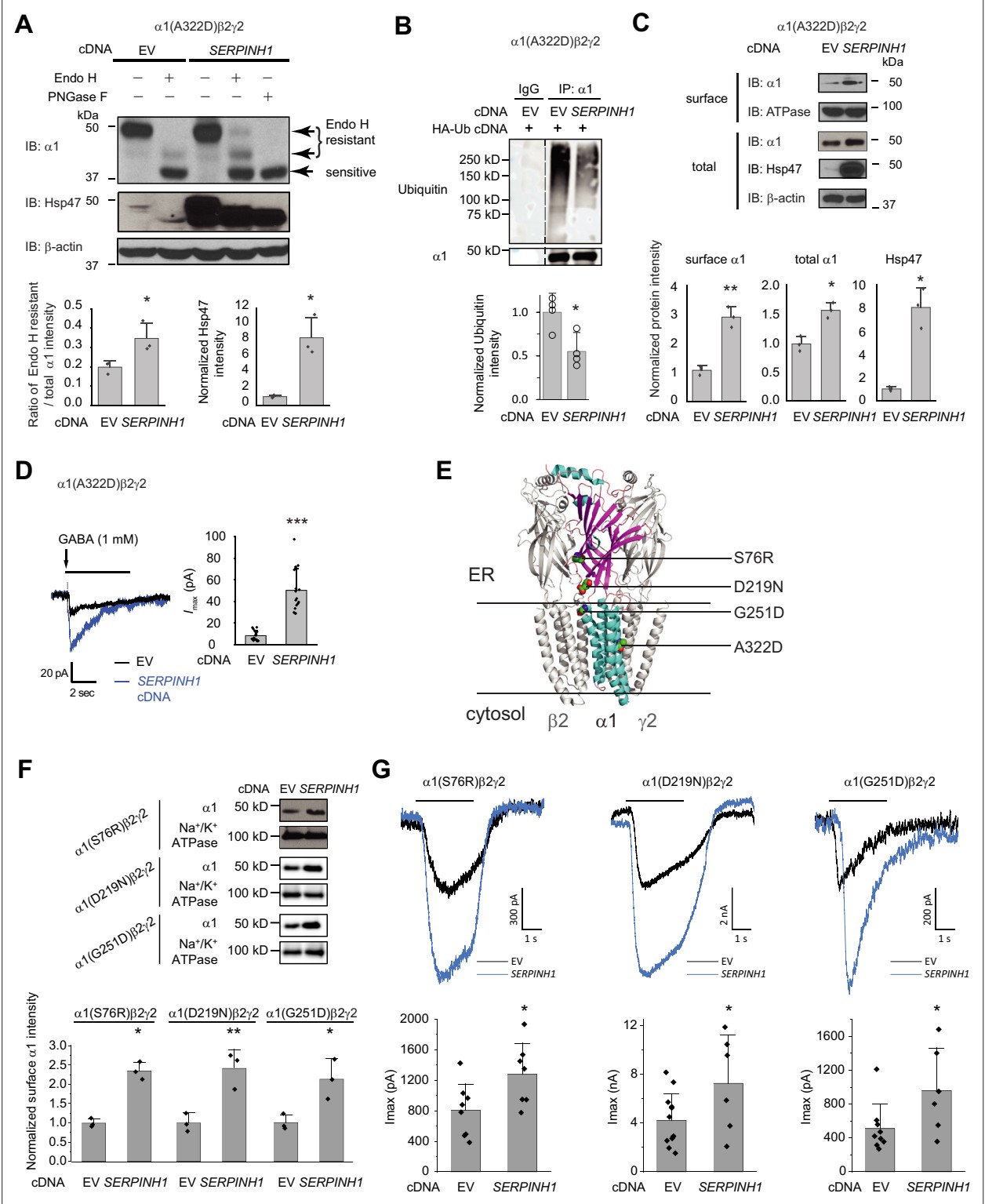

**Figure 5.** Hsp47 positively regulates the functional surface expression of epilepsy-associated GABA$_A$ receptors. (**A**) Overexpression of Hsp47 increases the endo H-resistant post-ER glycoform of the α1 subunit in HEK293T cells expressing α1(A322D)β2γ2 GABA$_A$ receptors. PNGase F treatment serves as a control for unglycosylated α1 subunit (lane 5). Two endo H-resistant bands were detected for the α1 subunit, indicated by the bracket (lanes 2 and 4). Three biological replicates were performed. Quantification of the ratio of endo H-resistant / total α1 subunit bands, as a measure of the ER-to-Golgi trafficking efficiency, is shown on the bottom. (**B**) HEK293T cells expressing α1(A322D)β2γ2 GABA$_A$ receptors were transfected with HA-ubiquitin together with empty vector (EV) control or *SERPINH1* cDNA plasmids. Forty-eight hours post transfection, cells were lysed and the total proteins

*Figure 5 continued on next page*

*Figure 5 continued*

were immunoprecipitated with anti-α1 antibody. The eluents were probed with indicated antibodies. Three biological replicates were performed. (**C**) HEK293T cells expressing α1(A322D)β2γ2 GABA$_A$ receptors were transfected with empty vector (EV) control or *SERPINH1* cDNA plasmids. Forty-eight hours post transfection, the surface proteins were measured using a cell surface protein biotinylation assay. The Na$^+$/K$^+$ ATPase serves as a loading control for biotinylated membrane proteins. Alternatively, cells were lysed, and the total cell lysates were subjected to reducing SDS-PAGE and immunoblotting analysis. β-actin serves as a total protein loading control. Three biological replicates were performed. Protein intensities were quantified using ImageJ and shown on the bottom. (**D**) Whole-cell patch clamping was performed to record GABA-induced currents. HEK293T cells were treated as in (**C**). The recording was carried out 48 hr post transfection. The holding potential was set at –60 mV. Representative traces were shown. Quantification of the peak currents ($I_{max}$) from 17 to 20 cells from three transfections is shown on the right. pA: picoampere. (**E**) Positions of the four α1 variants are displayed as space-filling models in the 3D structure of α1β2γ2 GABA$_A$ receptors, built from 6X3S.pdb using PyMOL. (**F**) HEK293T cells expressing α1(S76R)β2γ2, α1(D219N)β2γ2, or α1(G251D)β2γ2 GABA$_A$ receptors were transfected with EV control or *SERPINH1* cDNA plasmids. Forty-eight hours post transfection, the surface proteins were measured using a cell surface protein biotinylation assay. Three biological replicates were performed. (**G**) Whole-cell patch clamping was performed to record GABA-induced currents using the IonFlux Mercury 16 ensemble plates at a holding voltage of −60 mV. HEK293T cells were treated as in (**F**). The recording was carried out 48 hr post transfection. Application of GABA (100 μM, 3 s) is indicated by the horizontal bar above the current traces. Each ensemble recording enclosed 20 cells. Quantification of the peak currents ($I_{max}$) is shown on the bottom (n=6–12 ensembles). Each data point is reported as mean ± SD. Statistical significance was calculated using two-tailed Student's t-Test. *, p<0.05; **, p<0.01; ***, p<0.001.

The online version of this article includes the following source data and figure supplement(s) for figure 5:

**Source data 1.** Original files for the western blot analysis in *Figure 5A, B, C and F*.

**Source data 2.** PDF containing the original blots in *Figure 5* with the relevant bands clearly labeled.

**Source data 3.** Data used for graphs presented in *Figure 5A, B, C, D, F and G*.

**Figure supplement 1.** Effect of Hsp47 on the degradation of a GABA$_A$ receptor variant.

**Figure supplement 1—source data 1.** Original files for the western blot analysis in *Figure 5—figure supplement 1A,B,C*.

**Figure supplement 1—source data 2.** PDF containing the original blots in *Figure 5—figure supplement 1* with the relevant bands clearly labeled.

**Figure supplement 1—source data 3.** Data used for graphs presented in *Figure 5—figure supplement 1C*.

**Figure supplement 2.** Overexpression of Hsp47 in HEK293T cells expressing a variety of pathogenic GABA$_A$ receptor variants.

**Figure supplement 2—source data 1.** Original files for the western blot analysis in *Figure 5—figure supplement 2*.

**Figure supplement 2—source data 2.** PDF containing the original blots in *Figure 5—figure supplement 2* with the relevant bands clearly labeled.

**Figure supplement 2—source data 3.** Data used for graphs presented in *Figure 5—figure supplement 2*.

cells expressing α1(S76R)β2γ2 receptors, 1.72-fold in HEK293T cells expressing α1(D219N)β2γ2 GABA$_A$ receptors, and 1.87-fold in HEK293T cells expressing α1(G251D)β2γ2 GABA$_A$ receptors (*Figure 5G*), which are comparable to the peak currents for wild type receptors, suggesting the clinical potential of this approach. Furthermore, the effect of Hsp47 overexpression on increasing GABA-induced peak current amplitudes is more dramatic for trafficking-deficient α1 variants than for WT GABA$_A$ receptors (*Figure 3—figure supplement 1A*, *Figure 5D and G*).

## Depleting or overexpressing Hsp47 does not activate the UPR

Since the ER proteostasis network orchestrates the folding, assembly, degradation, trafficking of GABA$_A$ receptors (*Wang et al., 2022b*) and adapting the proteostasis network by activating the UPR, such as the ATF6 arm, rescues misfolding-prone GABA$_A$ variants (*Wang et al., 2022a*), we determined how genetic manipulations of Hsp47 influenced overall ER proteostasis. The UPR is the major cellular signaling pathway that monitors the ER proteostasis by using three arms, namely, the IRE1/XBP1s arm, ATF6 (activating transcription factor 6) arm, and PERK (protein kinase R-like ER kinase) arm (*Walter and Ron, 2011*). IRE1, ATF6, and PERK are all ER transmembrane proteins. IRE1 activation leads to its oligomerization and the splicing of XBP1 mRNA. Spliced XBP1 (XBP1s) is then translocated into the nucleus, acting as a transcription factor to regulate ER proteostasis. ATF6 activation leads to its translocation from the ER to the Golgi and the release of the active 50 kDa N-terminal fragment of ATF6 (ATF6-N), which will then be translocated to the nucleus to act as a transcription factor to enhance the ER folding capacity. PERK activation leads to its oligomerization and the ultimate induction of CHOP, a pro-apoptotic transcription factor. Knocking down Hsp47 (*Figure 6A*) or overexpressing Hsp47 (*Figure 6B*) did not change the protein levels of XBP1s, ATF6-N, or CHOP in HEK293T cells expressing WT α1β2γ2 or α1(A322D)β2γ2 GABA$_A$ receptors, whereas application of thapsigargin, a pan-UPR activator, increased the protein levels of XBP1s, ATF6-N, and CHOP significantly (*Figure 6B*).

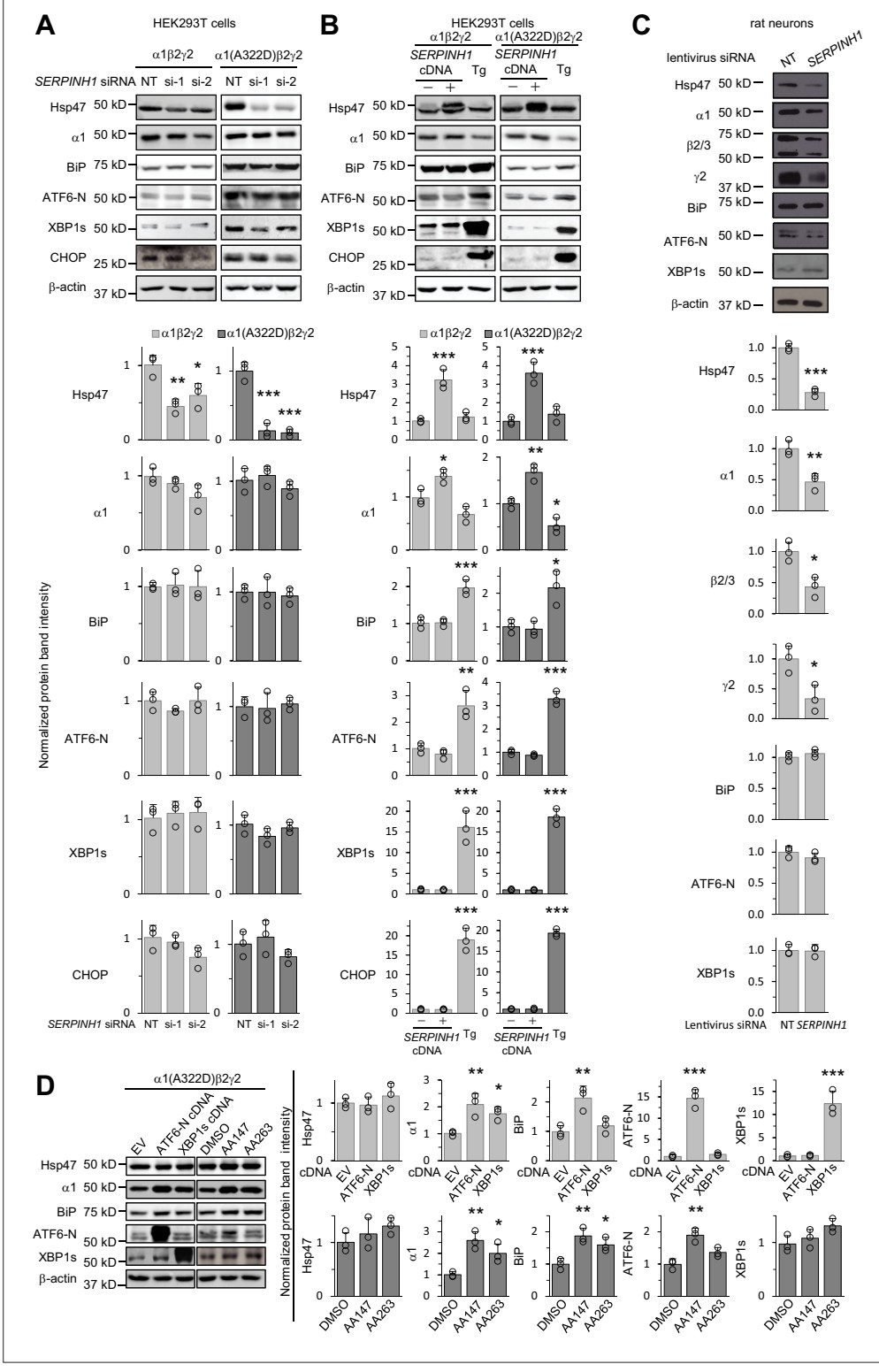

**Figure 6.** Genetic manipulations of Hsp47 do not activate the UPR. (**A**) HEK293T cells expressing WT α1β2γ2 or α1(A322D)β2γ2 GABA_A receptors were transiently transfected with non-targeting (NT) control siRNAs or siRNAs against *SERPINH1* (#1 or #2). Forty-eight hours post-transfection, cells were lysed for SDS-PAGE and Western blot analysis. (**B**) HEK293T cells expressing WT α1β2γ2 or α1(A322D)β2γ2 GABA_A receptors were transiently transfected with empty vector (EV) or *SERPINH1* cDNA plasmids. Forty-eight hours post-transfection, cells were lysed for SDS-PAGE and western blot analysis. Thapsigargin (Tg) (0.5 µM, 16 hr), a pan-UPR activator, was used as a positive

*Figure 6 continued on next page*

*Figure 6 continued*

control to induce the UPR. (**C**) Cultured cortical neurons from E18 rats were transduced with *SERPINH1* siRNA lentivirus or scrambled siRNA lentivirus at days in vitro (DIV) 10. Forty-eight hours post transduction, neurons were lysed for SDS-PAGE and western blot analysis. (**D**) HEK293T cells expressing α1(A322D)β2γ2 GABA$_A$ receptors were transiently transfected with empty vector (EV), ATF6-N cDNA, or XBP1s cDNA plasmids for 48 hr, or treated with DMSO vehicle control or ATF6 activators (AA147 (10 µM) or AA263 (10 µM)) for 24 hr. Afterwards, cells were lysed for SDS-PAGE and western blot analysis. Three biological replicates were performed. Each data point is reported as mean ± SD. Significant difference was analyzed by a one-way ANOVA followed by post hoc Tukey's HSD test (**A**, **B, D**) or t-test (**C**). *, $p<0.05$; **, $p<0.01$; ***, $p<0.001$.

The online version of this article includes the following source data for figure 6:

**Source data 1.** Original files for the western blot analysis in *Figure 6A, B, C and D*.

**Source data 2.** PDF containing the original blots in *Figure 6* with the relevant bands clearly labeled.

**Source data 3.** Data used for graphs presented in *Figure 6A, B, C and D*.

---

In addition, depleting or overexpressing Hsp47 did not influence the protein levels of BiP (*Figure 6A and B*), a prominent ATF6 downstream target (*Shoulders et al., 2013*). These results indicated that genetic operations of Hsp47 did not substantially induce the activation of the UPR in HEK293T cells. Moreover, we used primary cortical neurons to determine the effect of depleting Hsp47 on UPR activation and endogenous GABA$_A$ receptors. Clearly, knocking down Hsp47 reduced the protein levels of endogenous GABA$_A$ receptors, including the major α1, β2/β3, and γ2 subunits, without activating the UPR in cortical neurons (*Figure 6C*). Therefore, since genetic manipulations of Hsp47 did not induce the activation of the UPR, the positive effect of Hsp47 on enhancing the GABA$_A$ receptor assembly and trafficking was not likely through the alteration of the global ER proteostasis network.

Furthermore, we determined the effect of activating the UPR on the expression of Hsp47 in HEK239T cells expressing misfolding-prone GABA$_A$ receptors carrying the α1(A322D) variant. Activating the ATF6 arm genetically by overexpressing the active ATF6-N fragment or pharmacologically by the application of two ATF6 activators, namely AA147 and AA263 (*Plate et al., 2016*), did not significantly change Hsp47 protein levels (*Figure 6D*), whereas as expected, such operations increased α1(A322D) and BiP protein levels (*Figure 6D*; *Wang et al., 2022a*; *Fu et al., 2018*). In addition, activating the IRE1 arm genetically by overexpressing XBP1s did not increase Hsp47 protein levels (*Figure 6D*). These results indicated that activating the UPR did not change Hsp47 expression in HEK293T cells expressing a misfolding-prone GABA$_A$ receptor variant. Therefore, the effect of UPR activation on enhancing GABA$_A$ receptor proteostasis was not through Hsp47 upregulation.

## Hsp47 has a general role in increasing the surface expression of the Cys-loop receptors

The role of Hsp47 in regulating the maturation of ion channels has not been previously documented. We therefore expanded our investigation of the effect of Hsp47 to other members of the Cys-loop superfamily, to which GABA$_A$ receptors belong (*Changeux and Christopoulos, 2016*). The Cys-loop receptors, including nAChRs and serotonin type 3 receptors (5-HT$_3$Rs), are pentameric ligand-gated neuroreceptors, sharing a common structural scaffold, including a β-sheet-rich ER lumen domain (*Nemecz et al., 2016*; *Morales-Perez et al., 2016*). We chose to evaluate the effect of Hsp47 on nAChRs and 5-HT$_3$Rs. Heteropentameric α4β2 nAChRs and homopentameric α7 nAChRs are the major subtypes in the CNS (*Nashmi and Lester, 2006*). Overexpressing Hsp47 significantly increased the total protein levels of α4 and β2 subunits in HEK293T cells (*Figure 7A*). Previously, FRET experiments were developed to evaluate the assembly of α4β2 and α7 nAChRs (*Nashmi et al., 2003*; *Dau et al., 2013*; *Son et al., 2009*). Here, FRET assays demonstrated that overexpressing Hsp47 significantly increased the mean FRET efficiency of (CFP-β2)(YFP-α4) nAChRs from 27.7±16.4% to 40.9 ± 18.9% (*Figure 7B*), indicating that Hsp47 positively regulates the assembly of heteropentameric α4β2 receptors. In contrast, FRET experiments showed that Hsp47 overexpression did not influence the mean FRET efficiency of homopentameric α7 nAChRs (*Figure 7—figure supplement 1A*). In addition, Hsp47 overexpression did not change α7 total protein levels (*Figure 7—figure supplement 1B*). These results suggested that the capability of Hsp47 in the regulation of the biogenesis of α4β2 and α7 nAChRs is different. Furthermore, Hsp47 overexpression increased the nicotine-induced peak

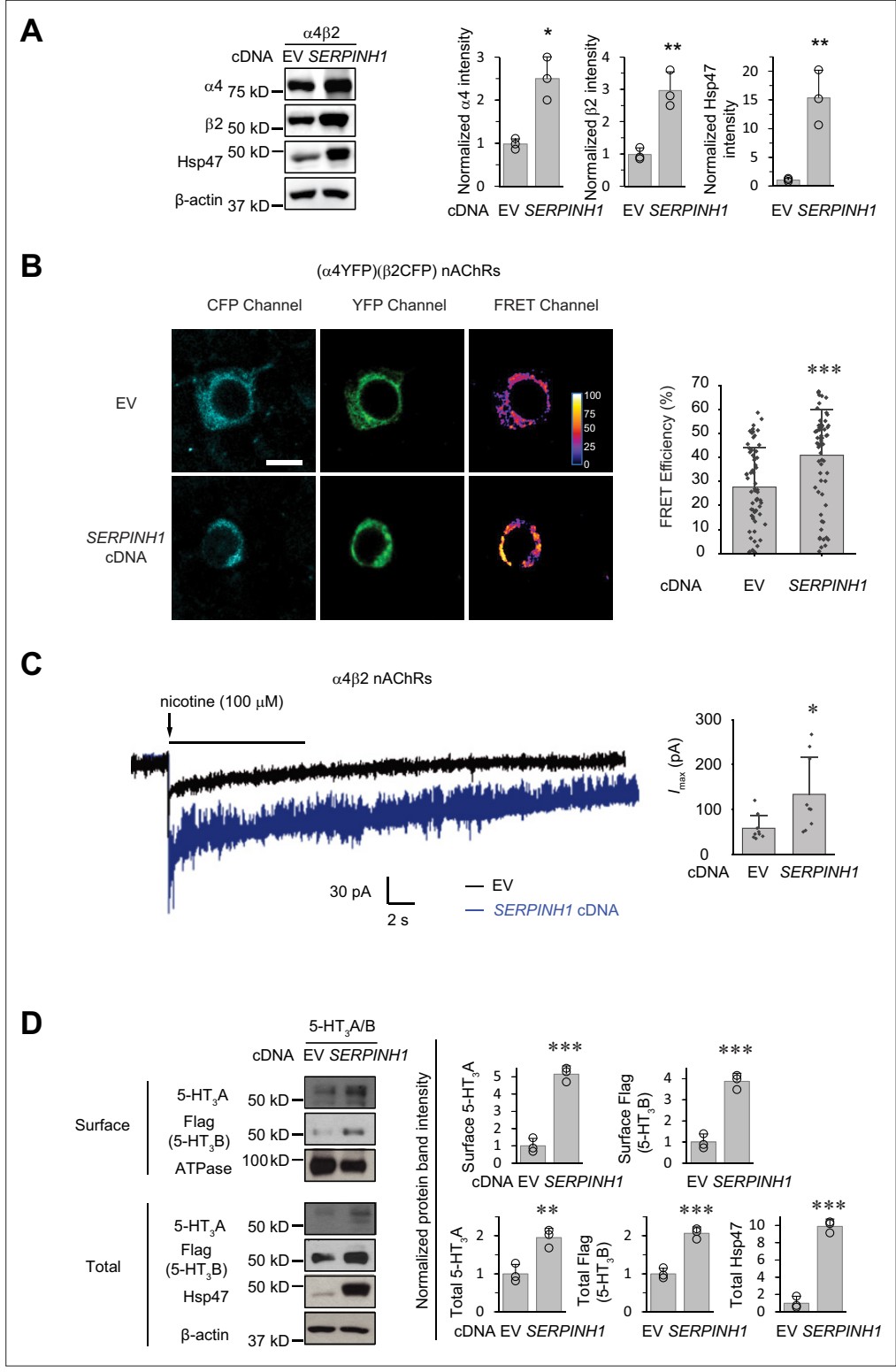

**Figure 7.** Hsp47 has a general effect on increasing the surface expression of the Cys-loop receptors. (**A**) HEK293T cells were transfected with nAChR subunits (α4 (*CHRNA4*) and β2 (*CHRNB2*)) and empty vector (EV) control or *SERPINH1* cDNA plasmids. Forty-eight hours post transfection, cells were lysed and the total proteins were evaluated using a western blot analysis. (**B**) Hsp47 overexpression increases FRET efficiency between CFP-tagged β2 subunit and YFP-tagged α4 subunit of nAChRs. HEK293T cells were transfected with CFP-tagged β2 subunit and YFP-tagged α4 subunit; in addition, cells were transfected with empty vector (EV) control or *SERPINH1* cDNA.

*Figure 7 continued on next page*

*Figure 7 continued*

Forty-eight hours post transfection, pixel-based FRET was used to measure the FRET efficiency between β2-CFP and α4-YFP by using a confocal microscope. Representative images were shown for the CFP channel (1st columns), YFP channel (2nd columns), and FRET efficiency (3rd columns). Scale bar = 10 μm. Quantification of the FRET efficiency from 60 to 70 cells from at least three transfections was achieved using the ImageJ PixFRET plug-in, and shown on the right. (**C**) HEK293T cells were transfected with CFP-tagged β2 subunit and YFP-tagged α4 subunit of nAChRs; in addition, cells were transfected with empty vector (EV) control or *SERPINH1* cDNA. Forty-eight hours post transfection, whole-cell patch clamping was performed to record nicotine-induced currents. Representative traces were shown. Quantification of the peak currents ($I_{max}$) from 9 cells from three transfections is shown on the right. The holding potential was set at –60 mV. pA: picoampere. (**D**) HEK293T cells were transfected with 5-HT$_3$R subunits (5-HT$_3$A and FLAG-tagged 5-HT$_3$B) and empty vector (EV) control or *SERPINH1* cDNA plasmids. Forty-eight hours post transfection, the surface proteins were measured using a cell surface protein biotinylation assay, and the total proteins were evaluated using a Western blot analysis. The Na$^+$/K$^+$ ATPase serves as a loading control for biotinylated membrane proteins. Three biological replicates were performed. Each data point is reported as mean ± SD. Statistical significance was calculated using two-tailed Student's t-Test. * p<0.05; ** p<0.01; *** p<0.001.

The online version of this article includes the following source data and figure supplement(s) for figure 7:

**Source data 1.** Original files for the western blot analysis in *Figure 7A and D*.

**Source data 2.** PDF containing the original blots in *Figure 7* with the relevant bands clearly labeled.

**Source data 3.** Data used for graphs presented in *Figure 7A, B, C and D*.

**Figure supplement 1.** Effect of Hsp47 on the biogenesis of α7 nAChRs.

**Figure supplement 1—source data 1.** Original files for the Western blot analysis in *Figure 7—figure supplement 1B*.

**Figure supplement 1—source data 2.** PDF containing the original blots in *Figure 7—figure supplement 1* with the relevant bands clearly labeled.

**Figure supplement 1—source data 3.** Data used for graphs presented in *Figure 7—figure supplement 1A, B*.

**Figure supplement 2.** Effect of Hsp47 on the biogenesis of structurally diverse proteins.

**Figure supplement 2—source data 1.** Original files for the western blot analysis in *Figure 7—figure supplement 2A,B,C*.

**Figure supplement 2—source data 2.** PDF containing the original blots in *Figure 7—figure supplement 2* with the relevant bands clearly labeled.

**Figure supplement 2—source data 3.** Data used for graphs presented in *Figure 7—figure supplement 2A,B,C*.

current from 57.9±28.6 pA (n=9) to 134±84 pA (n=9) in HEK293T cells expressing α4β2 nAChRs (*Figure 7C*). Therefore, Hsp47 positively regulates the assembly and thus function of α4β2 nAChRs.

In addition, 5-HT$_3$Rs are assembled from 5-HT$_3$A and 5-HT$_3$B subunits into heteropentamers in HEK293T cells (*Miles et al., 2013*). Clearly, overexpressing Hsp47 increased the surface expression of both the 5-HT$_3$A and 5-HT$_3$B subunits according to surface biotinylation assay as well as their total protein levels in HEK293T cells (*Figure 7D*). These results indicated that Hsp47 has a general role in promoting the surface presence of the heteropentameric Cys-loop receptors.

Furthermore, we tested the influence of Hsp47 on other ion channels that have different structural scaffolds compared to Cys-loop receptors, including tetrameric NMDA (N-methyl-D-aspartate) receptors and hERG potassium channels. NMDA receptors, assembled from two obligatory GluN1 subunits and two GluN2 subunits, play a critical role in mediating synaptic development and plasticity in the CNS (*Hansen et al., 2021*). Overexpressing Hsp47 did not change or decreased the total or surface protein levels of WT GluN2A subunits (*Figure 7—figure supplement 2A*, lane 2 to lane 1) and those of misfolding-prone M705V GluN2A subunits (*Zhang et al., 2024*; *Figure 7—figure supplement 2A*, lane 4 to lane 3) in HEK293T cells, indicating that Hsp47 did not positively regulate NMDA receptor proteostasis. Moreover, hERG potassium channels regulate cardiac action potential repolarization in the heart, and loss of their function leads to type 2 long QT syndrome (*Vandenberg et al., 2012*). Knocking down Hsp47 did not change the protein levels of the mature (*Figure 7—figure supplement 2B*, top 155 kDa bands) and immature (*Figure 7—figure supplement 2B*, bottom 135 kDa bands) protein levels of WT hERG as well as trafficking-deficient hERG variants (N470D and T65P) in HEK293T cells, indicating that Hsp47 did not influence biogenesis of hERG channels. Furthermore,

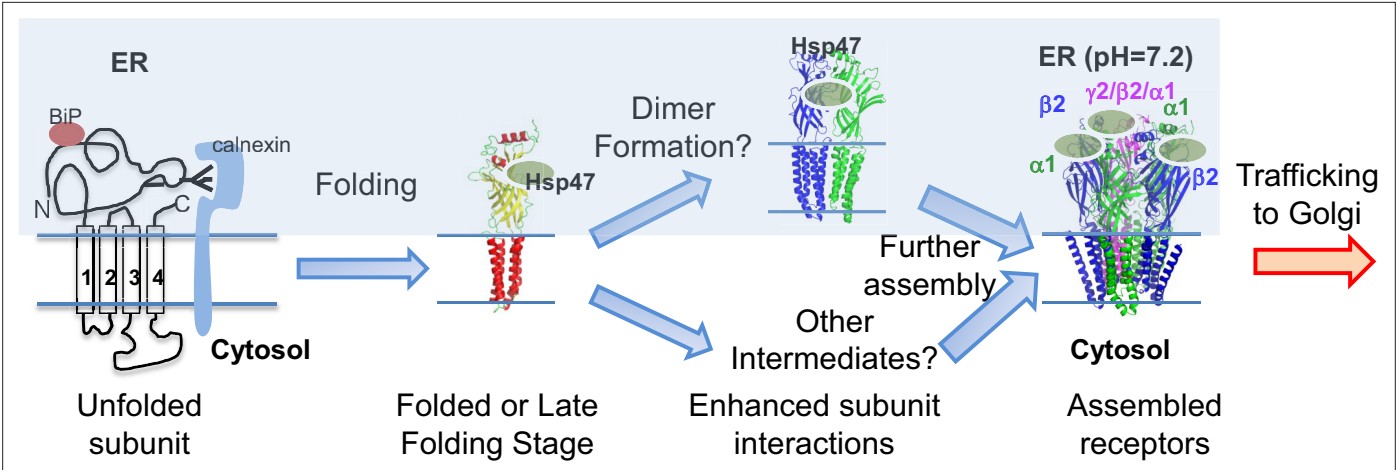

**Figure 8.** Proposed mechanism of Hsp47 in the assembly of GABA$_A$ receptors. BiP and calnexin assist the subunit folding early in the ER lumen. Hsp47 operates after BiP and binds the folded states of the α1 or β subunits in the ER lumen. Hsp47 links the α1 and β subunits and promotes their inter-subunit interactions. As a result, Hsp47 promotes the formation of assembly intermediates and the native pentameric receptors in the ER. Assembled receptors will traffic to the Golgi and onward to the plasma membrane for function.

we determined the effect of Hsp47 on an ERAD substrate, L444P β-glucocerebrosidase, the most common disease-causing variant in neuropathic Gaucher diseases (*Platt et al., 2018*). Knocking down Hsp47 did not change the total protein levels of folding-deficient L444P β-glucocerebrosidase in Gaucher patient-derived fibroblasts (*Figure 7—figure supplement 2C*), indicating that Hsp47 did not influence the biogenesis of an ERAD substrate in the ER lumen.

Therefore, our results indicated that Hsp47 had a general role in increasing the surface expression of heteropentameric Cys-loop receptors, including GABA$_A$ receptors, nAChRs, and 5-HT$_3$Rs. Furthermore, Hsp47 had certain selectivity for Cys-loop receptors since Hsp47 did not enhance the biogenesis of other structurally diverse ion channels, such as NMDA receptors and hERG channels, or an ERAD substrate in the ER lumen, L444P β-glucocerebrosidase. It is worth noting that it is expected that Hsp47's chaperoning role is not required for the assembly of hERG channels since they lack large ER luminal domains. Conversely, heterotetrameric NMDA receptors, which have large ER luminal domains, are more suitable substrates for the establishment of Hsp47's selectivity toward Cys-loop receptors.

## Discussion

*Figure 8* illustrates our proposed mechanism for Hsp47 positively regulating the surface trafficking of GABA$_A$ receptors. BiP and calnexin assist the subunit folding early in the ER lumen (*Di et al., 2013*). Hsp47 operates after BiP and binds the folded states or late folding stage of the α1 and β2 subunits in the ER lumen. Hsp47 binding enhances the interactions among adjacent subunits. As a result, Hsp47 promotes the formation of the assembly intermediates and the native pentameric receptors in the ER. It is unclear whether Hsp47 targets certain subunit-subunit interfaces, such as α1-β2, α1-γ2, or β2-γ2 interfaces. In addition, the assembly intermediates that Hsp47 promotes, such as dimers and trimers, require future investigation. Properly assembled receptors will traffic to the Golgi and onward to the plasma membrane for function. Regarding the impact of Hsp47 on GABA$_A$ receptor proteostasis, Hsp47 overexpression increases total protein levels of WT α1 (*Figure 4C*) and α1(A322D) proteins (*Figure 5C*), but does not change their apparent degradation rates (*Figure 3—figure supplement 1C*, *Figure 5—figure supplement 1A*). In addition, Hsp47 overexpressing enhances the ER-to-Golgi trafficking efficiency of WT α1 (*Figure 3A*) and α1(A322D) proteins (*Figure 5A*). Taken together, our data support that the critical effect of Hsp47 is due to the shift of GABA$_A$ receptor population toward the trafficking-competent states.

Our data demonstrated that knocking down Hsp47 (*Figure 6A and C*) or overexpressing Hsp47 (*Figure 6B*) does not activate IRE1/XBP1s, ATF6, or PERK pathway of the UPR in HEK293T cells expressing GABA$_A$ receptors or neurons. Therefore, the effect of Hsp47 on enhancing GABA$_A$ receptor

proteostasis is not likely through the alteration of the global ER proteostasis network. In addition, activating ATF6 or IRE1/XBP1s does not change Hsp47 protein levels in HEK293T cells expressing a misfolding-prone GABA$_A$ receptor variant (*Figure 6D*). Therefore, the effect of ATF6 or IRE1/XBP1s activation on enhancing GABA$_A$ receptor proteostasis is not through Hsp47 upregulation. Moreover, we showed that Hsp47 interacts with GABA$_A$ receptor subunits in the mouse brain (*Figure 1A*), in vitro (*Figure 1B*), and in cells (*Figure 3B and C*). Taken together, our data strongly support that the effect of Hsp47 on promoting GABA$_A$ receptor assembly and trafficking is direct.

In addition, we demonstrated that Hsp47 positively regulates the assembly and function of α4β2 nAChRs, but not the assembly of α7 nAChRs. Interestingly, it was reported that RIC-3, an ER trans-membrane protein, is important for the assembly of α7, but not α4β2 nAChRs (*Dau et al., 2013*; *Millar, 2008*), whereas NACHO (gene symbol: *TMEM35*), an ER transmembrane protein in the ER, can promote the assembly and function of both α4β2 and α7 nAChRs (*Gu et al., 2016*). Therefore, the differential role of chaperones on the assembly of various subtypes of multi-subunit ion channels requires further investigation. Nonetheless, it appears that Hsp47 acts as a chaperone that promotes the assembly process of heteropentameric Cys-loop receptors, including GABA$_A$ receptors, nAChRs, and 5-HT$_3$Rs. This supplements the canonical function of molecular chaperones that serve to assist protein folding and prevent protein aggregation. The assembly of GABA$_A$ receptor subunits is mediated by their N-terminal domains (*Taylor et al., 2000*; *Taylor et al., 1999*). For example, residues 86–96 within α1 subunits, especially Gln95, play an important role in their assembly with β3 subunits (*Taylor et al., 2000*). Here, we showed that an ER lumen localized chaperone, Hsp47, enhances the oligomerization of heteropentameric GABA$_A$ receptors.

Our results expand the client protein pool and function of Hsp47. Hsp47 was identified and is currently recognized as a collagen-specific chaperone (*Nagata et al., 1986*). Here, we demonstrate that Hsp47 has a general effect in proteostasis maintenance of GABA$_A$ receptor, nAChRs, and 5-HT$_3$Rs in the Cys-loop receptor superfamily. In addition, it was previously reported that Hsp47 physically interacts with amyloid precursor protein and regulates the secretion of Aβ-peptide (*Bianchi et al., 2011*). Therefore, Hsp47 could have diverse functions in the CNS. Here, we show that Hsp47 binds the ER luminal domain (ERD) of GABA$_A$ receptor α1 and β2 subunits with high-affinity. The GABA$_A$ receptor ERD domain is rich in β-sheets containing ten β-strands (*Miller and Aricescu, 2014*), whereas for its known substrate collagen, Hsp47 preferentially binds to the folded conformation of the collagen triple helices with a 2:1 stoichiometry (*Ono et al., 2012*; *Tasab et al., 2000*; *Widmer et al., 2012*). It is also known that Hsp47 interacts with the ERD of IRE1α, a major transducer of the UPR, with a Kd of 73.2±8.4 nM, to regulate the IRE1α oligomerization (*Sepulveda et al., 2018*); the ERD of IRE1α also adopts a β-sheet-rich structure with a triangular assembly of β-sheet clusters (*Zhou et al., 2006*). Therefore, it appears that Hsp47 can interact with both the β-sheet-rich structure and triple helix structure. How Hsp47 adopts these structurally diverse client proteins needs future investigations. As a pentameric channel, each GABA$_A$ receptor has five potential binding sites for Hsp47. However, each GABA$_A$ receptor only has two agonist binding sites in the N-terminal domain for GABA. Therefore, the binding stoichiometry between Hsp47 and the pentameric GABA$_A$ receptors merits future effort.

Recent advances in genetics identified mutations in GABA$_A$ receptors that are associated with idiopathic epilepsies (*Hernandez and Macdonald, 2019*; *Fu et al., 2022*; *El Achkar et al., 2021*). One mutation can compromise the receptor function by influencing the protein biogenesis pathways (transcription, translation, folding, assembly, trafficking, and endocytosis), ligand binding, channel gating, or their combinations. Recently, we showed that enhancing the ER folding capacity is a viable way to restore the surface expression and thus function of pathogenic GABA$_A$ receptors carrying misfolding-prone mutations in the α1 and γ2 subunits (*Wang et al., 2022a*; *Di et al., 2013*; *Han et al., 2015b*; *Fu et al., 2018*; *Di et al., 2021*; *Han et al., 2015a*). Numerous disease-causing mutations disrupt their folding and/or assembly, leading to reduced trafficking to the plasma membrane. These trafficking-deficient mutant subunits are retained in the ER and degraded by the ERAD pathway. Since we envisage that Hsp47 interacts with proteins in the folded states or late-folding states and stabilizes the assembly intermediates and assembled receptor complex, overexpressing Hsp47 can 'pull' more mutant subunits into the folded/assembled receptors that can engage the trafficking machinery for transport to the plasma membrane. Consequently, the function of the mutant receptors would be restored after Hsp47 overexpression. The rescuing mechanism of Hsp47 is similar to that of pharmacological chaperones, which bind to folded/assembled mutant GABA$_A$ receptors to stabilize their

conformation to enhance their surface transport (*Wang et al., 2023*; *Wang et al., 2014*). Indeed, we showed that Hsp47 overexpression as well as application of pharmacological chaperones, such as Hispidulin and TP003, enhanced the functional surface expression of a variety of trafficking-deficient α1 variants (*Figure 5D and G*; *Wang et al., 2023*). A promising therapeutic approach to treat genetic epilepsy resulting from GABA$_A$ receptor trafficking deficiency would be using small molecules to adapt the GABA$_A$ receptor proteostasis network, including Hsp47, to restore the surface trafficking and thus function of GABA$_A$ receptor variants. Therefore, this strategy serves as a proof-of-principle case for promoting the multi-subunit assembly process to ameliorate diseases resulting from membrane protein folding/assembly deficiencies.

## Materials and methods

### Lead Contact

Further information and requests for resources and reagents should be directed to and will be fulfilled by the Lead Contact, Ting-Wei Mu (tingwei.mu@case.edu).

### Materials availability

All plasmids generated in this study will be made available on request but we may request a completed Materials Transfer Agreement.

### Plasmids and siRNAs

The pCMV6 plasmids containing human GABA$_A$ receptor α1 subunit (*GABRA1*) (Uniprot #: P14867-1) (catalog #: RC205390), β2 subunit (*GABRB2*) (isoform 2, Uniprot #: P47870-1) (catalog #: RC216424), γ2 subunit (*GABRG2*) (isoform 2, Uniprot #: P18507-2) (catalog #: RC209260), 5-HT$_3$A subunit (catalog #: SC122578, *HTR3A*, NM_000869), C-terminal FLAG-tagged 5-HT$_3$B subunit (catalog #: MR206966, *HTR3B*, NM_020274), C-terminal FLAG-tagged hERG potassium channel (catalog #: RC215928, *KCNH2*, NM_000238, human), and pCMV6 Entry Vector plasmid (pCMV6-EV) (catalog #: PS100001) were obtained from Origene (Rockville, MD, USA). The pcDNA3.1-GluN2A (catalog # OHu24642D, *GRIN2A*, NM_000833, human) and pcDNA3.1-GluN1 (catalog # OHu22255D, *GRIN1*, NM_007327, human) plasmids were purchased from GenScript Biotech (Piscataway, NJ, USA). The nAChR α4 (*CHRNA4*) plasmid (Addgene plasmid # 24271; http://n2t.net/addgene:24271; RRID: Addgene_24271) and nAChR β2 (*CHRNB2*) plasmid were a gift from Henry Lester (Addgene plasmid # 24272; http://n2t.net/addgene:24272; RRID: Addgene_24272). The nAChR α7 (pcDNA3.1-*CHRNA7*) plasmid was a gift from Sherry Leonard & Henry Lester (Addgene plasmid # 62276; http://n2t.net/addgene:62276; RRID: Addgene_62276). The missense mutations, including S76R, D219N, G251D, A322D, C166A, or C166A/C180A in the GABA$_A$ receptor α1 subunit (*GABRA1*), M705V in GluN2A, and N470D or T65P in hERG (*KCNH2*), were constructed using a QuikChange II site-directed mutagenesis Kit (Agilent Genomics, catalog #: 200523). A FLAG tag was inserted between Leu31 and Gln32 of the α1 subunit (*GABRA1*) and between Asn28 and Asp29 of the β2 subunit of GABA$_A$ receptors (*GABRB2*) by using QuikChange II site-directed mutagenesis. For GABA$_A$ receptors, enhanced cyan fluorescent protein (CFP) was inserted between Lys364 and Asn365 in the TM3-TM4 intracellular loop of the α1 subunit (*GABRA1*), and enhanced yellow fluorescent proteins (YFP) was inserted between Lys359 and Met360 in the TM3-TM4 intracellular loop of the β2 subunit (*GABRB2*) by using the GenBuilder cloning kit (GenScript, catalog #: L00701). The construction of fluorescently tagged nAChR subunits were described previously (*Nashmi et al., 2003*; *Dau et al., 2013*): CFP was inserted into the TM3-TM4 intracellular loop of the β2 subunit (*CHRNB2*) (Addgene, catalog #: 15106), YFP was inserted into TM3-TM4 intracellular loop of the α4 subunit (*CHRNA4*) (Addgene, catalog #: 15245), and cerulean (a CFP variant) or venus (a YFP variant) was inserted into TM3-TM4 intracellular loop of the α7 subunit (*CHRNA7*). The human *SERPINH1* cDNA in pCMV6-XL5 plasmid was obtained from Origene (catalog#: SC119367). The pRK5-HA-Ubiquitin-WT was a gift from Ted Dawson (Addgene plasmid # 17608; http://n2t.net/addgene:17608; RRID:Addgene_17608).

ON-TARGETplus human *SERPINH1* siRNAs (catalog #: J-011230-05-0005 and J-011230-06-0005) and Non-Targeting negative control siRNAs (catalog #: D-001810-01-20) were purchased from Dharmacon. Scrambled siRNA GFP lentivector (catalog #: LV015-G) and *SERPINH1*-set of four siRNA lentivectors (rat) (catalog #: 435050960395) were obtained from Applied Biological Materials (BC,

Canada). psPAX2 (Addgene plasmid # 12260; http://n2t.net/addgene:12260; RRID:Addgene_12260) and pMD2.G (Addgene plasmid # 12259; http://n2t.net/addgene:12259; RRID:Addgene_12259) were a gift from Didier Trono. pCIG3 (pCMV-IRES-GFP version 3) was a gift from Felicia Goodrum (Addgene plasmid # 78264; http://n2t.net/addgene:78264; RRID:Addgene_78264). pHRIG-AktDN was a gift from Heng Zhao (Addgene plasmid # 53597; http://n2t.net/addgene:53597; RRID:Addgene_53597). To construct the pHRIG-*SERPINH1* ORF plasmid for lentiviral transduction, pHRIG-AktDN was digested with Sal1 and BamH1, and PCR-amplified *SERPINH1* ORF was sub-cloned using the Sal1 and BamH1 sites.

## Antibodies

The mouse monoclonal anti-GABA$_A$α1 subunit antibody (clone BD24, catalog #: MAB339), mouse monoclonal anti-GABA$_A$β2/3 subunit antibody (clone 62–3 G1, catalog #: 05–474), rabbit polyclonal anti-GABA$_A$β2 subunit antibody (catalog #: AB5561), rabbit polyclonal anti-GABA$_A$γ2 subunit antibody (catalog #: AB5559), and rabbit polyclonal anti-NeuN antibody (catalog #: ABN78) were obtained from Millipore (Burlington, MA). The rabbit polyclonal anti-GABA$_A$α1 antibody (catalog #: PPS022) was purchased from R&D systems (Minneapolis, MN). The goat polyclonal anti-GABA$_A$α1 subunit antibody (A-20; catalog #: SC-31405) and mouse monoclonal anti-nAChR α4 subunit (*CHRNA4*) antibody (catalog #: sc-74519) were obtained from Santa Cruz Biotechnology (Dallas, TX). The rabbit polyclonal anti-GABA$_A$α1 subunit antibody (catalog #: 224203) and rabbit polyclonal anti-GABA$_A$γ2 antibody (catalog #: 224003) were obtained from Synaptic Systems. The rabbit polyclonal anti-nAChR β2 subunit (*CHRNB2*) antibody (catalog #: 17844–1-AP), rabbit polyclonal anti-nAChR α7 subunit (*CHRNA7*) antibody (catalog #: 21379–1-AP), rabbit polyclonal anti-ATF6 antibody (catalog # 24169–1-AP), rabbit polyclonal anti-FLAG antibody (catalog #: 20543–1-AP), and mouse monoclonal anti-Hsp47 antibody (catalog #: 67863–1-lg) were purchased from Proteintech (Rosemont, IL, USA). The goat polyclonal anti-5HT$_3$A (catalog #: TA302602) antibody was obtained from Origene. The rabbit monoclonal anti-GluN2A antibody (catalog # ab124913), rabbit monoclonal anti-Hsp47 antibody (catalog #: ab109117), rabbit monoclonal anti-Grp78/BiP (*HSPA5*) antibody (catalog #: ab108613), and rabbit monoclonal anti-Na$^+$/K$^+$ ATPase (catalog #: ab76020) were obtained from Abcam (Waltham, MA). The rabbit polyclonal anti-hERG (*KCNH2*) antibody (catalog #: PA3-860) was obtained from Thermo Fisher. The rabbit monoclonal anti-XBP1s antibody (catalog # 12782 S), mouse monoclonal anti-CHOP antibody (catalog # 2895 S), and mouse monoclonal anti-His tag antibody (catalog #: 2366 S) were obtained from Cell Signaling (Danvers, MA, USA). The rabbit polyclonal anti-glucocerebrosidase antibody (catalog #: G4046), mouse monoclonal anti-FLAG antibody (catalog #: F1804), and anti-β-actin antibody (catalog #: A1978) came from Sigma (St. Louis, MO). The fluorescent anti-β-actin antibody Rhodamine came from Biorad (catalog #: 12004163). The rabbit polyclonal anti-Grp78/BiP (*HSPA5*) antibody (catalog # AP5041c) was obtained from Abgent (San Diego, CA, USA). The mouse monoclonal anti-Hsp47 antibody (catalog #: ADI-SPA-470-F) came from Enzo Life Sciences (Farmingdale, NY).

## Cell culture and transfection

HEK293T cells were obtained from ATCC (catalog #: CRL-3216, donor sex: female) or Abgent (catalog #: CL1032). Patient-derived skin fibroblasts harbouring L444P β-glucocerebrosidase were obtained from Coriell Institute (catalog #: GM20272, donor sex: male). No mycoplasma contamination was detected. Cells were maintained in Dulbecco's Modified Eagle Medium (DMEM) (Fisher Scientific, Waltham, MA, catalog #: 10–013-CV) with 10% heat-inactivated fetal bovine serum (Fisher Scientific, catalog #: SH30396.03HI) and 1% Penicillin-Streptomycin (Fisher Scientific, catalog #: SV30010) at 37 °C in 5% CO$_2$. Monolayers were passaged upon reaching confluency with 0.05% trypsin protease (Fisher Scientific, catalog #: SH30236.01). Cells were grown in 6-well plates or 10 cm dishes and allowed to reach ~70% confluency before transient transfection according to the manufacturer's instruction. For plasmid transfection, TransIT-2020 (Mirus Bio, Madison, WI, catalog #: MIR 5406) was used; for siRNA transfection, HiPerfect Transfection Reagent (QIAGEN, catalog #: 301707) was used with 50 nM siRNAs. HEK293T cells stably expressing α1β2γ2 or α1(A322D)β2γ2 GABA$_A$ receptors were generated using the G418 selectin method, as described previously (*Wang et al., 2022a*; *Fu et al., 2018*). Forty-eight hours post transfection, cells were harvested for protein analysis.

## Lentivirus transduction in rat neurons

Lentivirus production and transduction in neurons was performed as described previously (*Whittsette et al., 2022*). Briefly, HEK293T cells were transfected with a *SERPINH1*-set of four siRNA lentivectors (rat) (Applied Biological Materials, catalog #: 435050960395), scrambled siRNA GFP lentivector (Applied Biological Materials, catalog #: LV015-G), pCIG3 lentivector, or pHRIG- *SERPINH1* cDNA lentivector together with psPAX2 and pMD2.G plasmids using TransIT-2020. The medium was changed after 8 hr incubation, and cells were incubated for additional 36–48 hr. Then the medium was collected, filtered, and concentrated using Lenti-X concentrator (Takara Bio, Catalog #: 631231). The lentivirus was quantified with the qPCR lentivirus titration kit (Applied Biological Materials, catalog #: LV900), and stored at −80 °C.

Sprague Dawley rat E18 hippocampus (catalog #: SDEHP) and E18 cortex (catalog #: SDECX) were obtained from BrainBits (Springfield, IL). Neurons were isolated and cultured following the company's instruction. Briefly, tissues were digested with papain (2 mg/ml) (Sigma, catalog #: P4762) at 30 °C for 10 min and triturated with a fire-polished sterile glass pipette for 1 min. Neurons were then plated onto poly-D-lysine (PDL) (Sigma, catalog #: P6407) and Laminin (Sigma, catalog #: L2020)-coated glass coverslips in a 24-well plate. Neurons were maintained in neuronal culture media containing Neurobasal (Thermo Fisher, catalog #: 21103049), 2% B27 (Thermo Fisher, catalog #: 17504044), 0.5 mM GlutaMax (Thermo Fisher, catalog #: 35050061), and 1% penicillin-streptomycin (Thermo Fisher Scientific, catalog #: SV30010) at 37 °C in 5% $CO_2$. Neurons were subjected to transduction with lentivirus at days in vitro (DIV) 10, and immunofluorescence staining and electrophysiology were performed at DIV 12.

## Mouse brain homogenization

Male C57BL/6 J mice (Jackson Laboratory, RRID:IMSR_JAX:000664) at 8–10 weeks were sacrificed and the cortex was isolated and homogenized in the homogenization buffer (25 mM Tris, pH 7.6, 150 mM NaCl, 1 mM EDTA, and 2% Triton X-100) supplemented with the Roche complete protease inhibitor cocktail (Roche; catalog #: 4693159001). The homogenates were centrifuged at 800×*g* for 10 min at 4 °C, and the supernatants were collected. The pellet was re-homogenized in additional homogenization buffer and centrifuged at 800×*g* for 10 min at 4 °C. The supernatants were combined and rotated for 2 hr at 4 °C, and then centrifuged at 15,000×*g* for 30 min at 4 °C. The resulting supernatant was collected as mouse brain homogenate. Protein concentration was determined by a MicroBCA assay (Pierce, catalog #: 23235). This animal study (Protocol #: 2018–0017) was approved by the Institutional Animal Care and Use Committees (IACUC) at Case Western Reserve University and was carried out in agreement with the recommendation of the American Veterinary Medical Association Panel on Euthanasia. Animals were maintained in groups. The ARRIVE guidelines have been followed.

## Western blot analysis

Cells were harvested and lysed with lysis buffer (50 mM Tris, pH 7.5, 150 mM NaCl, and 1% Triton X-100) or RIPA buffer (50 mM Tris, pH 7.4, 150 mM NaCl, 5 mM EDTA, pH 8.0, 2% NP-40, 0.5% sodium deoxycholate, and 0.1% SDS) supplemented with Roche complete protease inhibitor cocktail. Lysates were cleared by centrifugation (20,000×*g*, 10 min, 4 °C). Protein concentration was determined by MicroBCA assay. Aliquots of cell lysates were separated in an 8% SDS-PAGE gel, and western blot analysis was performed using the appropriate antibodies. Band intensity was quantified using ImageJ software from the National Institute of Health (*Schneider et al., 2012*). For non-reducing protein gels, cell lysates were loaded in the Laemmli sample buffer (Bio-Rad, Hercules, CA, catalog #: 1610737); for reducing protein gels, cell lysates were loaded in the Laemmli sample buffer (Bio-Rad, catalog #: 1610737) supplemented with 100 mM dithiothreitol (DTT) to reduce the disulfide bonds. Endoglycosidase H (endo H) (New England Biolabs, catalog #: P0703L) enzyme digestion or Peptide-N-Glycosidase F (PNGase F) (New England Biolabs, Ipswich, MA, catalog #: P0704L) enzyme digestion was performed according to manufacturer's instruction and the published procedure (*Di et al., 2013*).

## Immunoprecipitation

Cell lysates (500 µg) or mouse brain homogenates (1 mg) were pre-cleared with 30 µL of protein A/G plus-agarose beads (Santa Cruz Biotechnology, catalog #: SC-2003) and 1.0 µg of normal mouse IgG (Santa Cruz Biotechnology, catalog #: SC-2025) for 1 hr at 4 °C to remove nonspecific binding proteins.

The pre-cleared samples were incubated with 2.0 µg of mouse anti-α1 antibody, mouse anti-Hsp47 antibody, or normal mouse IgG as a negative control for 1 hr at 4 °C and then with 30 µL of protein A/G plus agarose beads overnight at 4 °C. Afterward, the beads were collected by centrifugation at 8000×$g$ for 30 s, and washed three times with lysis buffer. The complex was eluted by incubation with 30 µL of Laemmli sample buffer loading buffer in the presence of 100 mM DTT. The immunopurified eluents were separated in an 8% SDS-PAGE gel, and western blot analysis was performed using appropriate antibodies.

## In vitro protein binding assay

One µg of GST epitope tag protein (GST) (Novus Biologicals, Centennial, CO, catalog #: NBC1-18537), GST-tagged human GABA$_A$ receptor α1 subunit protein (GST-α1) (Abnova, Walnut, CA, catalog #: H00002554-P01), or GST-tagged human GABA$_A$ receptor β2 subunit protein (GST-β2) (Abnova, catalog #: H00002561-P01) was mixed with 4 µg of recombinant His-tagged human Hsp47 protein (Novus, catalog #: NBC1-22576) in 500 µL of lysis buffer (50 mM Tris, pH 7.5, 150 mM NaCl, and 1% Triton X-100). The protein complex was isolated by immunoprecipitation using a mouse anti-His antibody (Cell Signaling, catalog #: 2366 S) followed by SDS-PAGE and Western blot analysis with a rabbit anti-GABA$_A$ receptor α1 subunit antibody (R&D Systems, catalog #: PPS022), a rabbit anti-GABA$_A$ receptor β2 subunit antibody (Millipore, catalog #: AB5561), or a mouse anti-His antibody.

In addition, 4 µg of recombinant His-tagged human Hsp47 protein (Novus, catalog #: NBC1-22576) was mixed with 1 µg of FLAG-tagged ZIP7 (Origene, catalog #: TP313722), 1 µg of hERG (Abonva, catalog #: H00003757-G01), or 1 µg of GST-tagged human GABA$_A$ receptor α1 subunit protein (GST-α1) (Abnova, Walnut, CA, catalog #: H00002554-P01) in 500 µL of binding buffer (50 mM Tris, pH 7.5, 150 mM NaCl, and 2 mM N-dodecyl-β-D-maltoside (DDM)). The protein complex was isolated by immunoprecipitation using a mouse anti-His antibody (Cell Signaling, catalog #: 2366 S) followed by SDS-PAGE and Western blot analysis with a rabbit anti-GABA$_A$ receptor α1 subunit antibody (R&D Systems, catalog #: PS022), a rabbit anti-FLAG antibody (Proteintech, catalog #: 20543–1-AP), a rabbit anti-hERG antibody (Thermo Fisher, catalog #: PA3-860), or a mouse anti-His antibody.

## MicroScale thermophoresis (MST)

MST experiments were carried out to measure the binding affinity between the ER luminal domain of human GABA$_A$ receptor α1 subunits (α1-ERD) or ECD of human GABA$_A$ receptor β2 subunits (β2-ERD) and an ER luminal chaperone, Hsp47, using a Monolith NT.115 instrument (NanoTemper Technologies Inc, South San Francisco, CA). Monolith His-Tag Labeling Kit RED-tris-NTA 2$^{nd}$ Generation (Nano-Temper Technologies, catalog #: MO-L018) was used to label recombinant His-α1-ERD (MyBioSource, San Diego, CA, catalog #: MBS948971) and recombinant His-β2-ERD (MyBioSource, San Diego, CA, catalog #: MBS953526). One hundred µL of 200 nM RED-tris-NTA dye in PBST buffer (137 mM NaCl, 2.7 mM KCl, 10 mM Na$_2$HPO$_4$, 1.8 mM KH$_2$PO$_4$, pH 7.4, 0.05% Tween-20) was mixed with 100 µL of 200 nM His-α1-ERD or His-β2-ERD in PBST, and the reaction mixture was incubated for 30 min at room temperature in the dark. The serial dilutions of the ligand proteins, recombinant human Hsp47 (Abcam, catalog #: ab86918) (0.2 nM to 10 µM) in PBST, were prepared in Maxymum Recovery PCR tubes (Axygen, Union City, CA, catalog #: PCR-02-L-C) with a final volume of 5 µL in each tube. Then 5 µL of 100 nM RED labeled His-α1-ERD or His-β2-ERD was added to each PCR tube containing 5 µL of the ligand proteins. The samples were loaded to Monolith Premium Capillaries (NanoTemper Technologies, catalog #: MO-K025) and measured using a Monolith NT.115 instrument with the settings of 40% LED/excitation and 40% MST power. The data were collected and analyzed using the Monolith software. Ligand-dependent changes in temperature-related intensity change (TRIC) are plotted as F$_{norm}$ (normalized fluorescence) values vs. ligand concentrations in a dose-response curve for the calculation of the dissociation constant (Kd).

## Circular dichroism (CD) measurements

CD experiments were carried out using a JASCO J-1500 spectrophotometer with a 1 mm pathlength quartz cuvette at room temperature. The protein samples were diluted to 0.15 mg/mL in Dulbecco's phosphate buffered saline (DPBS; Thermo Fisher Scientific, catalog #: SH3002803). Each CD Spectrum was measured by accumulating three spectra to obtain the average with the blank correction. Data was analyzed using the JASCO Spectra Manager software (Version 2).

## Biotinylation of cell surface proteins

Cells were plated in 6 cm dishes for surface biotinylation experiments according to the published procedure (*Di et al., 2013*). Briefly, intact cells were washed twice with ice-cold Dulbecco's phosphate buffered saline (DPBS) (Fisher Scientific, catalog #: SH3002803). To label surface membrane proteins, cells were incubated with the membrane-impermeable biotinylation reagent Sulfo-NHS SS-Biotin (0.5 mg/mL; Pierce, catalog #: 21331) in DPBS containing 0.1 mM $CaCl_2$ and 1 mM $MgCl_2$ (DPBS +CM) for 30 min at 4 °C. To quench the reaction, cells were incubated with 10 mM glycine in ice-cold DPBS +CM twice for 5 min at 4 °C. Sulfhydryl groups were blocked by incubating the cells with 5 nM N-ethylmaleimide (NEM; Pierce, catalog #: 23030) in DPBS for 15 min at room temperature. Cells were then solubilized for 1 h at 4 °C in solubilization buffer (50 mM Tris–HCl, 150 mM NaCl, 5 mM EDTA, pH 7.5, 1% Triton X-100) supplemented with Roche complete protease inhibitor cocktail and 5 mM NEM. The samples were centrifuged at 20,000×*g* for 10 min at 4 °C to pellet cellular debris. The supernatant contained the biotinylated surface proteins. The concentration of the supernatant was measured using microBCA assay. Biotinylated surface proteins were affinity-purified from the above supernatant by incubating for 1 hr at 4 °C with 100 µL of immobilized neutravidin-conjugated agarose bead slurry (Pierce, catalog #: 29200). The samples were then centrifuged at 20,000×*g* for 10 min at 4 °C. The beads were washed six times with solubilization buffer. Surface proteins were eluted from beads by boiling for 5 min with 200 µL of LSB/Urea buffer (2 x Laemmli sample buffer (LSB) with 100 mM DTT and 6 M urea; pH 6.8) for SDS-PAGE and Western blotting analysis.

## Immunofluorescence staining and confocal microscopy

Neuron staining and confocal immunofluorescence microscopy analysis were performed as described previously (*Di et al., 2013*; *Whittsette et al., 2022*). Briefly, to label cell surface proteins, primary neurons on coverslips were fixed with 2% paraformaldehyde in DPBS for 10 min. We then blocked with 10% goat serum (Thermo Fisher, catalog #: 16210064) in DPBS for 0.5 hr, and without detergent permeabilization, incubated with 100 µL of appropriate primary antibodies against the $GABA_A$ receptor α1 subunit (Synaptic Systems, Goettingen, Germany, catalog #: 224203) (1:250 dilution), β2/3 subunits (Millipore, catalog #: 05–474) (1:250 dilution), or γ2 subunit (Synaptic Systems, Goettingen, Germany, catalog #: 224003) (1:250 dilution), diluted in 2% goat serum in DPBS, at room temperature for 1 hr. Then the neurons were incubated with Alexa 594-conjugated goat anti-rabbit antibody (Thermo Fisher, catalog #: A11037), or Alexa 594-conjugated goat anti-mouse antibody (Thermo Fisher, catalog #: A11032) (1:500 dilution) diluted in 2% goat serum in DPBS for 1 hr. Afterward, cells were permeabilized with saponin (0.2%) for 5 min and incubated with DAPI (1 µg/mL) (Thermo Fisher, catalog #: D1306) for 3 min to stain the nucleus. To label intracellular proteins, neurons were fixed with 4% paraformaldehyde in DPBS for 15 min, permeabilized with saponin (0.2%) in DPBS for 15 min, and blocked with 10% goat serum in DPBS for 0.5 hr at room temperature. Then neurons were labeled with appropriate primary antibodies against Hsp47 (Enzo Life Sciences, catalog #: ADI-SPA-470-F (1:100 dilution) or Abcam, catalog #: ab109117 (1:250 dilution)) or NeuN, a neuron nuclei marker (Millipore, catalog #: ABN78) (1:500 dilution), diluted in 2% goat serum in DPBS, for 1 hr. The neurons were then incubated with Alexa 594-conjugated goat anti-mouse antibody (Thermo Fisher, catalog #: A11032) (1:500 dilution), or Alexa 405-conjugated goat anti-rabbit antibody (ThermoFisher, catalog #: A31556) (1:500 dilution), diluted in 2% goat serum in DPBS, for 1 h. The coverslips were then mounted using fluoromount-G (VWR, catalog #: 100502–406) and sealed. An Olympus IX-81 Fluoview FV1000 confocal laser scanning system was used for imaging with a 60×objective by using FV10-ASW software. The images were analyzed using ImageJ software (*Schneider et al., 2012*).

## Pixel-based sensitized acceptor emission FRET microscopy

Pixel-by-pixel based sensitized acceptor FRET microscopy was performed as described previously (*Dau et al., 2013*; *Moss et al., 2009*). For FRET experiments on $GABA_A$ receptors, (1) for FRET pair samples, HEK293T cells on coverslips were transfected with α1-CFP (donor) (0.5 µg), β2-YFP (acceptor) (0.5 µg), and γ2 (0.5 µg) subunits; (2) for the donor-only samples, to determine the spectral bleed-through (SBT) parameter for the donor, HEK293T cells were transfected with α1-CFP (donor) (0.5 µg), β2 (0.5 µg), and γ2 (0.5 µg) subunits; (3) for the acceptor-only samples, to determine the SBT parameter for the acceptor, HEK293T cells were transfected with α1 (0.5 µg), β2-YFP (acceptor) (0.5 µg), and γ2 (0.5 µg) subunits. For FRET experiments on nAChRs, HEK293T cells were transfected

with β2-CFP (donor) (0.7 μg) and α4-YFP (acceptor) (0.7 μg) or α7-cerulean (donor) (0.7 μg) and α7-venus (acceptor) (0.7 μg); in addition, the donor-only samples or the acceptor-only samples were prepared to determine the SBT parameters for the donor or the acceptor, respectively. Whole-cell patch-clamp recordings in HEK293T cells showed that fluorescently tagged ion channels have similar peak current amplitudes and dose-response curves to the agonists as compared to untagged ion channels (*Figure 4—figure supplement 1*; *Dau et al., 2013*). The coverslips were then mounted using fluoromount-G and sealed. An Olympus Fluoview FV1000 confocal laser scanning system was used for imaging with a 60×1.35 numerical aperture oil objective by using Olympus FV10-ASW software.

For the FRET pair samples, donor images were acquired at an excitation wavelength of 433 nm and an emission wavelength of 478 nm, FRET images at 433 nm excitation and 528 nm emission wavelengths, and acceptor images at 514 nm excitation and 528 nm emission wavelengths. For the donor-only samples, donor images were acquired at 433 nm excitation and 478 nm emission wavelengths, and FRET images at 433 nm excitation and 528 nm emission wavelengths. For the acceptor-only samples, FRET images were acquired at an excitation of 433 nm and an emission of 528 nm, and acceptor images at an excitation of 514 nm and an emission of 528 nm. Image analysis of FRET efficiencies was performed using the PixFRET plugin of the ImageJ software (*Feige et al., 2005*). The bleed-through was determined for the donor and the acceptor. With the background and bleed-through correction, the net FRET (nFRET) was calculated according to *equation (1)*.

$$nFRET = I_{FRET} - SBT_{donor} * I_{donor} - SBT_{acceptor} * I_{acceptor} \tag{1}$$

FRET efficiencies from sensitized emission experiments were calculated according to *equation (2)*.

$$E_{FRET} = 1 - I_{DA}/I_D \tag{2}$$

$E_{FRET}$ represents FRET efficiency, $I_{DA}$ represents the emission intensity of the donor in the presence of the acceptor, and $I_D$ represents the emission intensity of the donor alone. Since $I_D$ can be estimated by adding the nFRET signal amplitude to the amplitude of $I_{DA}$ (*Elangovan et al., 2003*), FRET efficiency was calculated according to *equation (3)*.

$$E_{PRET} = 1 - (I_{DA}/(I_{DA} + nFRET)) \tag{3}$$

## Whole-cell patch-clamp electrophysiology

Whole-cell patch-clamp recording was performed at room temperature, as described previously for GABA$_A$ receptors (*Han et al., 2015a*) or nAChRs (*Henderson et al., 2016*). Briefly, the glass electrodes have a tip resistance of 3–5 MΩ when filled with intracellular solution. For GABA$_A$ receptor recording, the intracellular solution was composed of (in mM): 153 KCl, 1 MgCl$_2$, 5 EGTA, 10 HEPES, and 5 Mg-ATP (adjusted to pH 7.3 with KOH); the extracellular solution was composed of (in mM): 142 NaCl, 8 KCl, 6 MgCl$_2$, 1 CaCl$_2$, 10 glucose, 10 HEPES, and 120 nM fenvalerate (adjusted to pH 7.4 with NaOH). For nAChR recording, the intracellular solution was composed of (in mM): 135 K-gluconate, 5 KCl, 5 EGTA, 0.5 CaCl$_2$, 10 HEPES, 2 Mg-ATP, and 0.1 GTP (adjusted to pH 7.2 with Tris-base); the extracellular solution was composed of (in mM): 140 NaCl, 5 KCl, 2 CaCl$_2$, 1 MgCl$_2$, 10 HEPES, and 10 glucose (adjusted to pH 7.3 with Tris-base). For GABA$_A$ receptor recordings, coverslips containing cells were placed in a RC-25 recording chamber (Warner Instruments) on the stage of an Olympus IX-71 inverted fluorescence microscope and perfused with extracellular solution. Fast chemical application was accomplished with a pressure-controlled perfusion system (Warner Instruments) positioned within 50 μm of the cell utilizing a Quartz MicroManifold with 100 μm inner diameter inlet tubes (ALA Scientific). The whole-cell currents were recorded at a holding potential of −60 mV in voltage-clamp mode using an Axopatch 200B amplifier (Molecular Devices, San Jose, CA). The signals were acquired at 10 kHz and filtered at 2 kHz using pClamp10 software (Molecular Devices). For nAChR recordings, cells were visualized with an upright microscope (Axio Examiner A1, Zeiss) equipped with an Axiocam 702 mono camera. Whole-cell currents were recorded at a holding potential of −60 mV in voltage clamp mode using an Integrated Patch-Clamp Amplifier (Sutter). Signals were detected at 10 kHz and filtered at 2 kHz using SutterPatch acquisition software.

## Automated patch-clamping with IonFlux Mercury 16 instrument

Automated patch clamping electrophysiology was performed in HEK293T cells expressing $GABA_A$ receptors on the Ionflux Mercury 16 instrument (Fluxion Biosciences, California), as previously described (*Wang et al., 2023*). Briefly, on the day of experiments, cells were detached using accutase (Sigma Aldrich, catalog #: A6964-500mL) and suspended in serum free medium HEK293 SFM II (Gibco, catalog #: 11686–029), supplemented with 25 mM HEPES (Gibco, catalog #: 15630–080) and 1% penicillin streptomycin. Then cells were pelleted, resuspended in the extracellular solution, and added to the Ionflux ensemble plate 16 (Fluxion Biosciences, catalog #: 910–0054). Each ensemble recording enclosed 20 cells. Whole-cell GABA-induced currents were recorded at a holding potential of –60 mV. The signals were acquired and analyzed by Fluxion Data Analyzer.

## Quantification and statistical analysis

All data are presented as mean ± SD. If two groups were compared, statistical significance was calculated using an unpaired Student's t-test; if more than two groups were compared, we used ANOVA followed by post hoc Tukey. A $p<0.05$ was considered statistically significant. *, $p<0.05$; **, $p<0.01$; ***, $p<0.001$.

## Acknowledgements

This work was supported by the National Institutes of Health (R01NS105789 and R01NS117176 to TM). We thank Dr. Matthias Buck (Case Western Reserve University, Cleveland, Ohio) for the help from his group on the MST experiments and Dr. Yinghua Chen (Case Western Reserve University, Cleveland, Ohio) for her assistance with the Circular Dichroism experiments.

## Additional information

### Funding

| Funder | Grant reference number | Author |
| --- | --- | --- |
| National Institute of Neurological Disorders and Stroke | R01NS105789 | Ting-Wei Mu |
| National Institute of Neurological Disorders and Stroke | R01NS117176 | Ting-Wei Mu |

The funders had no role in study design, data collection and interpretation, or the decision to submit the work for publication.

### Author contributions

Ya-Juan Wang, Conceptualization, Data curation, Formal analysis, Writing - original draft, Writing - review and editing; Xiao-Jing Di, Data curation, Formal analysis, Writing - original draft, Writing - review and editing; Pei-Pei Zhang, Xi Chen, Raad Nashmi, Brandon J Henderson, Fraser J Moss, Data curation, Formal analysis, Writing - review and editing; Marnie P Williams, Data curation, Writing - review and editing; Dong-Yun Han, Data curation, Formal analysis; Ting-Wei Mu, Conceptualization, Formal analysis, Supervision, Funding acquisition, Writing - original draft, Writing - review and editing

### Author ORCIDs

Fraser J Moss ⓘ http://orcid.org/0000-0002-8519-6991
Ting-Wei Mu ⓘ http://orcid.org/0000-0002-6419-9296

### Ethics

This animal study (Protocol #: 2018-0017) was approved by the Institutional Animal Care and Use Committees (IACUC) at Case Western Reserve University and was carried out in agreement with the recommendation of the American Veterinary Medical Association Panel on Euthanasia. Animals were maintained in groups. The ARRIVE guidelines have been followed.

Decision letter and Author response
Decision letter https://doi.org/10.7554/eLife.84798.sa1
Author response https://doi.org/10.7554/eLife.84798.sa2

## Additional files

### Supplementary files
• MDAR checklist

### Data availability
This paper does not report original code. All source data are available in this paper and supplementary information.

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

# Appendix 1

**Appendix 1—key resources table**

| Reagent type (species) or resource | Designation | Source or reference | Identifiers | Additional information |
|---|---|---|---|---|
| Cell line (*Homo-sapiens*) | HEK293T | ATCC | Cat#: CRL-3216 | |
| Cell line (*Homo-sapiens*) | HEK293T | Abgent | Cat#: CL1032 | |
| Cell line (*Homo-sapiens*) | Fibroblasts harbouring L444P β-glucocerebrosidase | Coriell Institute | Cat#: GM20272 | |
| Transfected construct (Human) | siRNA to *SERPINH1* | Dharmacon | Cat#: J-011230-05-0005 | |
| Transfected construct (Human) | siRNA to *SERPINH1* | Dharmacon | Cat#: J-011230-06-0005 | |
| Transfected construct (Human) | siRNA, non-targeting control | Dharmacon | Cat#: D-001810-01-20 | |
| Biological sample (rat) | E18 hippocampus | BrainBits | Cat#: SDEHP | |
| Biological sample (rat) | E18 cortex | BrainBits | Cat#: SDECX | |
| Antibody | anti-GABA$_A$α1 (mouse monoclonal) | Millipore | Cat#: MAB339 | WB (1:2000) |
| Antibody | anti-GABA$_A$β2/3 (mouse monoclonal) | Millipore | Cat#: 05–474 | WB (1:1000) IF (1:250) |
| Antibody | anti-GABA$_A$β2 (rabbit polyclonal) | Millipore | Cat#: AB5561 | WB (1:1000) |
| Antibody | anti-GABA$_A$γ2 (rabbit polyclonal) | Millipore | Cat#: AB5559 | WB (1:1000) |
| Antibody | anti-NeuN (rabbit polyclonal) | Millipore | Cat#: ABN78 | IF (1:500) |
| Antibody | anti-GABA$_A$α1 (rabbit polyclonal) | R&D systems | Cat#: PPS022 | WB (1:1000) |
| Antibody | anti-GABA$_A$α1 (goat polyclonal) | Santa Cruz Biotechnology | Cat#: SC-31405 | WB (1:1000) |
| Antibody | anti-nAChR α4 (mouse monoclonal) | Santa Cruz Biotechnology | Cat#: sc-74519 | WB (1:1000) |
| Antibody | anti-GABA$_A$α1 (rabbit polyclonal) | Synaptic Systems | Cat#: 224203 | IF (1:250) |
| Antibody | anti-GABA$_A$γ2 (rabbit polyclonal) | Synaptic Systems | Cat#: 224003 | IF (1:250) |
| Antibody | anti-nAChR β2 (rabbit polyclonal) | Proteintech | Cat#: 17844–1-AP | WB (1:1000) |
| Antibody | anti-nAChR α7 (rabbit polyclonal) | Proteintech | Cat#: 21379–1-AP | WB (1:1000) |
| Antibody | anti-ATF6 (rabbit polyclonal) | Proteintech | Cat#: 24169–1-AP | WB (1:2000) |
| Antibody | anti-FLAG (rabbit polyclonal) | Proteintech | Cat#: 20543–1-AP | WB (1:2000) |
| Antibody | anti-Hsp47 (mouse monoclonal) | Proteintech | Cat#: 67863–1-Ig | WB (1:1000) |
| Antibody | anti-5-HT$_3$A (goat polyclonal) | Origene | Cat#: TA302602 | WB (1:1000) |
| Antibody | anti-GluN2A (rabbit monoclonal) | Abcam | Cat#: ab124913 | WB (1:3000) |
| Antibody | anti-Hsp47 (rabbit monoclonal) | Abcam | Cat#: ab109117 | WB (1:1000) IF: (1:250) |
| Antibody | anti-Grp78 (rabbit monoclonal) | Abcam | Cat#: ab108613 | WB (1:2000) |
| Antibody | anti-Na+/K+ATPase (rabbit monoclonal) | Abcam | Cat#: ab76020 | WB (1:10,000) |
| Antibody | anti-hERG (rabbit polyclonal) | ThermoFisher | Cat#: PA3-860 | WB (1:10,000) |
| Antibody | anti-XBP1s (rabbit monoclonal) | Cell Signaling | Cat#: 12782 S | WB (1:1000) |
| Antibody | anti-CHOP (mouse monoclonal) | Cell Signaling | Cat#: 2895 S | WB (1:1000) |
| Antibody | anti-His (mouse monoclonal) | Cell Signaling | Cat#: 2366 S | WB (1:1000) |

*Appendix 1 Continued on next page*

*Appendix 1 Continued*

| Reagent type (species) or resource | Designation | Source or reference | Identifiers | Additional information |
|---|---|---|---|---|
| Antibody | anti- glucocerbrosidase (rabbit polyclonal) | Sigma | Cat#: G4046 | WB (1:1000) |
| Antibody | anti-FLAG (mouse monoclonal) | Sigma | Cat#: F1804 | WB (1:2000) |
| Antibody | anti-β-actin (mouse monoclonal) | Sigma | Cat#: A1978 | WB (1:20,000) |
| Antibody | Fluorescent anti-β-actin Rhodamine | Biorad | Cat#: 12004163 | WB (1:10,000) |
| Antibody | anti- Grp78 (rabbit polyclonal) | Abgent | Cat#: AP5041c | WB (1:5000) |
| Antibody | anti-Hsp47 (mouse monoclonal) | Enzo Life Sciences | Cat#: ADI-SPA-470-F | WB (1:2000) IF (1:250) |
| Antibody | Alexa 594-conjugated goat anti-rabbit | ThermoFisher | Cat#: A11037 | IF (1:500) |
| Antibody | Alexa 594-conjugated goat anti-mouse | ThermoFisher | Cat#: A11032 | IF (1:500) |
| Antibody | Alexa 405-conjugated goat anti-rabbit | ThermoFisher | Cat#: A31556 | IF (1:500) |
| Recombinant DNA reagent | pCMV6-*GABRA1*-CFP (plasmid) | This paper | | See Materials and Methods, Section Plasmids and siRNAs |
| Recombinant DNA reagent | pCMV6-*GABRB2* (plasmid) | Origene | Cat#: RC216424 | |
| Recombinant DNA reagent | pCMV6-*GABRB2*-YFP (plasmid) | This paper | | See Materials and Methods, Section Plasmids and siRNAs |
| Recombinant DNA reagent | pCMV6-*GABRG2* (plasmid) | Origene | Cat#: RC209260 | |
| Recombinant DNA reagent | pCMV6-*HTR3A* (plasmid) | Origene | Cat#: SC122578 | |
| Recombinant DNA reagent | pCMV6-*HTR3B* (plasmid) | Origene | Cat#: MR206966 | |
| Recombinant DNA reagent | pCMV6-*KCNH2* (plasmid) | Origene | Cat#: RC215928 | |
| Recombinant DNA reagent | pCMV6-Entry Vector (plasmid) | Origene | Cat#: PS100001 | |
| Recombinant DNA reagent | pcDNA3.1-*GRIN2A* (plasmid) | GenScript | Cat#: OHu24642D | |
| Recombinant DNA reagent | pcDNA3.1-*GRIN1* (plasmid) | GenScript | Cat#: OHu22255D | |
| Recombinant DNA reagent | *CHRNA4* (plasmid) | Addgene | RRID: Addgene_24271 | |
| Recombinant DNA reagent | *CHRNA4*-YFP (plasmid) | Addgene | RRID: Addgene_15245 | |
| Recombinant DNA reagent | *CHRNB2* (plasmid) | Addgene | RRID: Addgene_24272 | |
| Recombinant DNA reagent | *CHRNB2*-CFP (plasmid) | Addgene | RRID: Addgene_15106 | |
| Recombinant DNA reagent | pcDNA3.1-*CHRNA7* (plasmid) | Addgene | RRID: Addgene_62276 | |
| Recombinant DNA reagent | pcDNA3.1-*CHRNA7*-cerulean (plasmid) | https://pubmed.ncbi.nlm.nih.gov/23586521/ | | |
| Recombinant DNA reagent | pcDNA3.1-*CHRNA7*-venus (plasmid) | https://pubmed.ncbi.nlm.nih.gov/23586521/ | | |
| Recombinant DNA reagent | pCMV6-XL5-*SERPINH1* (plasmid) | Origene | Cat#: SC119367 | |
| Recombinant DNA reagent | pRK5-HA-Ubiquitin (plasmid) | Addgene | RRID: Addgene_17608 | |
| Recombinant DNA reagent | psPAX2 (plasmid) | Addgene | RRID: Addgene_12260 | |
| Recombinant DNA reagent | pMD2.G (plasmid) | Addgene | RRID: Addgene_12259 | |

*Appendix 1 Continued on next page*

*Appendix 1 Continued*

| Reagent type (species) or resource | Designation | Source or reference | Identifiers | Additional information |
|---|---|---|---|---|
| Recombinant DNA reagent | pCIG3 (plasmid) | Addgene | RRID: Addgene_78264 | |
| Recombinant DNA reagent | pHRIG-AktDN (plasmid) | Addgene | RRID: Addgene_53597 | |
| Recombinant DNA reagent | pHRIG-*SERPINH1* (plasmid) | This paper | | See Materials and Methods, Section Plasmids and siRNAs |
| Recombinant DNA reagent | Scrambled siRNA GFP lentivector (plasmid) | Applied Biological Materials | Cat#: LV015-G | |
| Recombinant DNA reagent | *SERPINH1* siRNA lentivector (smartpool) (plasmid) | Applied Biological Materials | Cat#: 435050960395 | |
| Peptide, recombinant protein | GST | Novus | Cat#: NBC1-18537 | |
| Peptide, recombinant protein | GST-GABRA1 | Abnova | Cat#: H00002554-P01 | |
| Peptide, recombinant protein | GST-GABRB2 | Abnova | Cat#: H00002561-P01 | |
| Peptide, recombinant protein | His-Hsp47 | Novus | Cat#: NBC1-22576 | |
| Peptide, recombinant protein | FLAG-ZIP7 | Origene | Cat#: TP313722 | |
| Peptide, recombinant protein | hERG | Abnova | Cat#: H00003757-G01 | |
| Peptide, recombinant protein | His-GABRA1-ERD | MyBioSource | Cat#: MBS948971 | |
| Peptide, recombinant protein | His-GABRB2-ERD | MyBioSource | Cat#: MBS953526 | |
| Peptide, recombinant protein | Hsp47 | Abcam | Cat#: ab86918 | |
| Commercial assay or kit | GenBuilder cloning kit | GenScript | Cat#: L00701 | |
| Chemical compound, drug | AA147 | Tocris Bioscience | Cat#: 6759 | |
| Chemical compound, drug | AA263 | Sigma | Cat#: R699470 | |
| Chemical compound, drug | DAPI | ThermoFisher | Cat#: D1306 | |
| Software, algorithm | ImageJ | National Institutes of Health | https://imagej.nih.gov/ij/ | |
| Software, algorithm | pClamp10 | Molecular Devices | https://www.moleculardevices.com/products/axon-patch-clamp-system | |
| Software, algorithm | Automatic patch clamping | Ionflux Mercury16 | https://www.fluxionbio.com/ionflux-mercury-automated-patch-clamp | |
| Software, algorithm | JASCO Spectra Manager | JASCO | | |
| Software, algorithm | Monolith NT.115 | NanoTemper | | |
| Software, algorithm | Origin | Origin Lab | https://www.originlab.com/ | |
| Software, algorithm | PyMOL | Schrodinger | https://pymol.org/ | |

