## [Editor Report]

This important study defines new functions for the ER-resident protein HSP47 in the quality control of multi-pass membrane receptor proteins. The evidence supporting the conclusions is solid, with rigorous biochemical assays employed in appropriate models. However, additional consideration regarding the mechanism of HSP47-dependent regulation of membrane protein quality control would have strengthened the study. This work will be of broad interest to cell biologists and biochemists interested in the fields of proteostasis, membrane protein quality control, and neuroreceptor signaling.

---

## [Decision Letter]

**Decision letter after peer review:**

Thank you for submitting your article "Hsp47 Promotes Biogenesis of Multi-subunit Neuroreceptors in the Endoplasmic Reticulum" for consideration by *eLife*. Your article has been reviewed by 3 peer reviewers, one of whom is a member of our Board of Reviewing Editors, and the evaluation has been overseen by David Ron as the Senior Editor. The reviewers have opted to remain anonymous.

Essential revisions (for the authors):

All three reviewers were enthusiastic about this manuscript and the evidence supporting a new role for HSP47 in promoting the assembly and trafficking of membrane protein receptors including GABAA. However, there were two main points that came up during the review of your manuscript that we feel would be important to address in a revision, both of which relate to the mechanism of HSP47's role in GABAA trafficking:

1. There are questions regarding the directness of the effect of HSP47 on GABAA folding and trafficking. While it is clear from the evidence presented that HSP47 binds to GABAA subunits, the specific importance of these interactions on changes to GABAA assembly and trafficking was not entirely clear. For example, does genetic manipulation of GABAA subunits globally alter ER proteostasis through mechanisms such as UPR activation? This type of global ER remodeling could indirectly impact GABAA receptor assembly and trafficking. In our reviews, we suggest a number of experiments that could help clarify this point, which we feel would strengthen the overall findings.

2. The second point centers on the proposed mechanism by which HSP47 is proposed to influence GABAA ER quality control and trafficking. In the manuscript, the authors propose that HSP47 binds to folded GABAA subunits after BiP-mediated chaperoning to facilitate assembly. However, there are questions related to the impact of HSP47 on GABAA ER proteostasis, specifically whether HSP47 leads to GABAA ER accumulation or degradation. Additional clarification on the specific impact of HSP47 on GABAA within the ER would also improve the overall manuscript in the opinion of the reviewers.

We encourage you to address these, and other comments brought up in the below reviews, in follow-up work to continue to define the important role of HSP47 in the folding, assembly, and trafficking of Cys-loop receptors such as GABAA.

*Reviewer #1 (Recommendations for the authors):*

1. As indicated above, an important unanswered question is 'how does HSP47 genetic manipulation influence overall ER proteostasis and function in neurons and HEK293T cells?' While it's clear that HSP47 directly binds GABAA, it isn't clear that the observed phenotypes can be attributed to this binding. For example, does knockdown or overexpression of HSP47 influence stress-responsive signaling pathways such as the UPR in the presence of GABAA overexpression? It is known that HSP47 regulates IRE1 signaling and the authors have previously demonstrated that altered UPR signaling can influence GABAA trafficking and activity. The authors should monitor the expression of ER stress-responsive genes to determine the potential impact of depletion/overexpression of HSP47 on ER function.

2. Along the same lines, do stress-induced increases in HSP47 influence the assembly, trafficking, and activity of GABAA receptors in the context of a UPR? In other words, does increase HSP47 contribute to improved ER proteostasis of GABAA receptors observed upon the activation of UPR signaling pathways?

3. It would also be useful to show that ER proteostasis (i.e., folding, trafficking) of another protein is unaffected by HSP47 overexpression. This doesn't necessarily need to be a membrane protein but could give support to the idea that the observed effects are selective for Cys loop membrane proteins.

4. It wasn't clear to me how the BiP binding to the GABAA subunit described in Figure 1C was performed. Was this in the presence of nucleotide? How can one compare the relative binding affinities of BiP and HSP47 without considering the nucleotide-dependent binding affinity of BiP. I'd more be careful with that comparison in the text.

5. The authors convincingly demonstrate the importance of HSP47 in promoting the assembly of the GABAA receptors, but it isn't clear if HSP47 specifically is important for the assembly of a single interface or if all receptor interfaces require HSP47 for assembly. In other words, if HSP47 is depleted/overexpressed are specific assembly intermediates identified? It isn't clear from Figure 4C, although higher levels of HSP47 do appear to increase a band in the alpha1 blot (not observed in the beta2 blot) of ~120 kDa that could reflect a specific interaction.

6. With the mutant GABAA subunit, the authors show increased trafficking and activity in response to HSP47 overexpression. How does this improved trafficking/activity compare to the wild-type protein?

*Reviewer #2 (Recommendations for the authors):*

Wang and colleagues provide evidence that the ER-resident HSP47 chaperone promotes the folding of GABA receptor subunits and the assembly of GABA subunits into multimeric ion channels. They demonstrate HSP47 can rescue the folding and function of a missense mutant A332D epilepsy-associated GABA subunit. They also demonstrate similar enhanced folding/function for acetylcholine receptor assembly. Overall, the experimental data in mouse brain lysates, cultured rat neurons, and HEK293 cells are well-presented and provide insight into new ion channel clients of the HSP47 chaperone and suggest that HSP47 expression can target disease-associated ion channel mutations.

1. Authors show in Figure S1 model that unassembled or misfolded GABA subunits are degraded. Is HSP47 knock-down leading to degradation of GABA subunits, either for wild-type or the A332D mutant subunit, or are the GABA subunits trapped in a BiP-bound state without HSP47? Conversely, is over-expression reducing ubiquitination and ERAD of the GABA subunits?

2. Authors show that HSP47 enhances folding and assembly for wild-type and A332D mutant GABA subunits. Authors show that, in the case of A332D, HSP47 overexpression enhances ion channel function. What about wild-type GABA subunits? Are there functional consequences for HSP47 overexpression on wild-type channel ion conductance, since they show improved folding and assembly?

3. Authors propose that HSP47 preferentially binds the correctly folded conformation of GABA subunits. But for the A332D mutation, how can it adopt a correctly folded conformation given the inherent amino acid substitution? Can authors provide more insight into the consequences and A332D on GABA, and how HSP47 is overcoming the mutation?

4. The functional studies of HSP47 overexpression are performed in HEK293 cells. Is it feasible to evaluate HSP47 overexpression in the native rat neurons on GABA ion channel conductance?

*Reviewer #3 (Recommendations for the authors):*

– In Figure 1B, it would be good to include a control of Hsp47 pull-down with another membrane protein to indicate specificity.

– Overall, very little alpha1 appears to be EndoH resistant under EV conditions in HEK293 cells. Does this low trafficking efficiency track with rat primary neurons or brain tissue more broadly?

– Can the authors explain what the middle band is between EndoH resistant and sensitive (Figure 3A, 5A)?

– For Figure 4B and 5A, a blot should be added to quantify Hsp47 overexpression.

– Please provide additional details for the FRET measurements of the homopentameric alpha7 nAChR complex assembly. Since alpha7 subunits are tagged with a mixture of fluorophores this would produce heterogenous complex mixtures. This should lower the sensitivity of the assay to detect changes in the assembly efficiency. How does the LOD change relative to heteromeric complexes?

---

## [Author Response]

Essential revisions (for the authors):All three reviewers were enthusiastic about this manuscript and the evidence supporting a new role for HSP47 in promoting the assembly and trafficking of membrane protein receptors including GABAA. However, there were two main points that came up during the review of your manuscript that we feel would be important to address in a revision, both of which relate to the mechanism of HSP47's role in GABAA trafficking:1. There are questions regarding the directness of the effect of HSP47 on GABAA folding and trafficking. While it is clear from the evidence presented that HSP47 binds to GABAA subunits, the specific importance of these interactions on changes to GABAA assembly and trafficking was not entirely clear. For example, does genetic manipulation of GABAA subunits globally alter ER proteostasis through mechanisms such as UPR activation? This type of global ER remodeling could indirectly impact GABAA receptor assembly and trafficking. In our reviews, we suggest a number of experiments that could help clarify this point, which we feel would strengthen the overall findings.

According to the reviewers’ comments, we carried out new experiments (Figure 6A-D) to determine the relationship between Hsp47 and UPR in the context of GABAA receptors. We summarized these results on pages 16-18 under a new subtitle “Depleting or overexpressing Hsp47 does not activate the UPR”.

In addition, on pages 21-22, in the discussion, we added, “Our data demonstrated that knocking down Hsp47 (Figure 6A, 6C) or overexpressing Hsp47 (Figure 6B) does not activate IRE1/XBP1s, ATF6, or PERK pathway of the UPR in HEK293T cells expressing GABAA receptors or neurons. Therefore, the effect of Hsp47 on enhancing GABAA receptor proteostasis is not likely through the alteration of the global ER proteostasis network. In addition, activating ATF6 or IRE1/XBP1s does not change Hsp47 protein levels in HEK293T cells expressing a misfolding-prone GABAA receptor variant (Figure 6D). Therefore, the effect of ATF6 or IRE1/XBP1s activation on enhancing GABAA receptor proteostasis is not through Hsp47 upregulation. Moreover, we showed that Hsp47 interacts with GABAA receptor subunits in the mouse brain (Figure 1A), in vitro (Figure 1B), and in cells (Figure 3B, 3C). Taken together, our data strongly support that the effect of Hsp47 on promoting GABAA receptor assembly and trafficking is direct.”

2. The second point centers on the proposed mechanism by which HSP47 is proposed to influence GABAA ER quality control and trafficking. In the manuscript, the authors propose that HSP47 binds to folded GABAA subunits after BiP-mediated chaperoning to facilitate assembly. However, there are questions related to the impact of HSP47 on GABAA ER proteostasis, specifically whether HSP47 leads to GABAA ER accumulation or degradation. Additional clarification on the specific impact of HSP47 on GABAA within the ER would also improve the overall manuscript in the opinion of the reviewers.

According to the reviewers’ comments, we carried out new experiments to determine the impact of Hsp47 on GABAA receptor accumulation and degradation (Figure 3—figure supplement 1 and Figure 5—figure supplement 1).

On page 21, in the discussion, we added, “Regarding the impact of Hsp47 on GABAA receptor proteostasis, Hsp47 overexpression increases total protein levels of WT α1 (Figure 4C) and α1(A322D) proteins (Figure 5C), but does not change their apparent degradation rates (Figure 3—figure supplement 1C, Figure 5—figure supplement 1A). In addition, Hsp47 overexpressing enhances the ER-to-Golgi trafficking efficiency of WT α1 (Figure 3A) and α1(A322D) proteins (Figure 5A). Taken together, our data support that the critical effect of Hsp47 is due to the shift of GABAA receptor population toward the trafficking-competent states.”

The mechanism of Hsp47 on GABAA receptors is similar to that of pharmacological chaperones. On page 24, in the discussion, we added, “Since we envisage that Hsp47 interacts with proteins in the folded states or late-folding states and stabilizes the assembly intermediates and assembled receptor complex, overexpressing Hsp47 can “pull” more mutant subunits into the folded/assembled receptors that can engage the trafficking machinery for transport to the plasma membrane. Consequently, the function of the mutant receptors would be restored after Hsp47 overexpression. The rescuing mechanism of Hsp47 is similar to that of pharmacological chaperones, which bind to folded/assembled mutant GABAA receptors to stabilize their conformation to enhance their surface transport [65,90]. Indeed, we showed that Hsp47 overexpression as well as application of pharmacological chaperones, such as Hispidulin and TP003, enhanced the functional surface expression of a variety of trafficking-deficient α1 variants (Figure 5D, 5G) [65]. Therefore, this strategy serves as a proof-of-principle case for promoting the multi-subunit assembly process to ameliorate diseases resulting from membrane protein folding/assembly deficiencies.”

In addition, we performed new experiments to determine the effect of Hsp47 on various ion channels and ERAD substrates (Figure 5E-G, Figure 7D, and Figure 7—figure supplement 2).

Our new experiments (Figure 5E-G) demonstrated that overexpressing Hsp47 rescues functional surface expressions of a number of trafficking-deficient, epilepsy-associated GABAA receptor variants, including α1(S76R), α1(D219N), and α1(G251D). Please see page 16 in the manuscript for details.

Our new experiments (Figure 7D, see page 19 for details) showed that Hsp47 overexpression increased the surface proteins of serotonin type 3 receptors, another member of Cys-loop receptors. Given Hsp47’s positive effects on heteropentameric GABAA receptors (Figure 2) and nAChRs (Figure 7A-C), it appears that Hsp47 has a general role in increasing the surface expression of heteropenatmeric Cys-loop receptors.

Moreover, we tested the influence of Hsp47 on other ion channels that have different structural scaffolds compared to Cys-loop receptors, including NMDA receptors and hERG potassium channels, and an ERAD substrate, L444P β-glucocerebrosidase. New results demonstrated that Hsp47 did not positively regulate the protein levels of NMDA receptors (Figure 7—figure supplement 2A), hERG channels (Figure 7—figure supplement 2B), or L444P β-glucocerebrosidase (Figure 7—figure supplement 2C) (see pages 19-20 for detail).

Taken together, on page 20, we stated, “Therefore, our results indicated that Hsp47 had a general role in increasing the surface expression of heteropentameric Cys-loop receptors, including GABAA receptors, nAChRs, and 5-HT_3_Rs. Furthermore, Hsp47 had certain selectivity for Cys-loop receptors since Hsp47 did not enhance the biogenesis of other structurally diverse ion channels, such as NMDA receptors and hERG channels, or an ERAD substrate in the ER lumen, L444P β-glucocerebrosidase.”

We encourage you to address these, and other comments brought up in the below reviews, in follow-up work to continue to define the important role of HSP47 in the folding, assembly, and trafficking of Cys-loop receptors such as GABAA.

We addressed each comment below.

Reviewer #1 (Recommendations for the authors):1. As indicated above, an important unanswered question is 'how does HSP47 genetic manipulation influence overall ER proteostasis and function in neurons and HEK293T cells?' While it's clear that HSP47 directly binds GABAA, it isn't clear that the observed phenotypes can be attributed to this binding. For example, does knockdown or overexpression of HSP47 influence stress-responsive signaling pathways such as the UPR in the presence of GABAA overexpression? It is known that HSP47 regulates IRE1 signaling and the authors have previously demonstrated that altered UPR signaling can influence GABAA trafficking and activity. The authors should monitor the expression of ER stress-responsive genes to determine the potential impact of depletion/overexpression of HSP47 on ER function.

We carried out new experiments to determine how genetic operations of Hsp47 influenced the UPR. Our results demonstrated that depleting Hsp47 or overexpressing Hsp47 did not activate the three arms of the UPR in HEK293T cells expressing WT α1β2γ2 or α1(A322D)β2γ2 GABAA receptors (Figure 6A, Figure 6B). In addition, depleting Hsp47 did not activate the UPR in primary cortical neurons (Figure 6C). Therefore, it appears that genetic manipulations of Hsp47 did not change the ER proteostasis environment globally, and thus the effect of Hsp47 on enhancing GABAA receptor assembly and trafficking is not likely an indirect effect through the UPR activation.

On page 16-18, we stated, “Depleting or overexpressing Hsp47 does not activate the UPR.

Since the ER proteostasis network orchestrates the folding, assembly, degradation, trafficking of GABAA receptors [40] and adapting the proteostasis network by activating the UPR, such as the ATF6 arm, rescues misfolding-prone GABAA variants [15], we determined how genetic manipulations of Hsp47 influenced overall ER proteostasis. The UPR is the major cellular signaling pathway that monitors the ER proteostasis by using three arms, namely, the IRE1/XBP1s arm, ATF6 (activating transcription factor 6) arm, and PERK (protein kinase R-like ER kinase) arm [66]. IRE1, ATF6, and PERK are all ER transmembrane proteins. IRE1 activation leads to its oligomerization and the splicing of XBP1 mRNA. Spliced XBP1 (XBP1s) is then translocated into the nucleus, acting as a transcription factor to regulate ER proteostasis. ATF6 activation leads to its translocation from the ER to the Golgi and the release of the active 50 kDa N-terminal fragment of ATF6 (ATF6-N), which will then be translocated to the nucleus to act as a transcription factor to enhance the ER folding capacity. PERK activation leads to its oligomerization and the ultimate induction of CHOP, a pro-apoptotic transcription factor. Knocking down Hsp47 (Figure 6A) or overexpressing Hsp47 (Figure 6B) did not change the protein levels of XBP1s, ATF6-N, or CHOP in HEK293T cells expressing WT α1β2γ2 or α1(A322D)β2γ2 GABAA receptors, whereas application of thapsigargin, a pan-UPR activator, increased the protein levels of XBP1s, ATF6-N, and CHOP significantly (Figure 6B). In addition, depleting or overexpressing Hsp47 did not influence the protein levels of BiP (Figure 6A, 6B), a prominent ATF6 downstream target [67]. These results indicated that genetic operations of Hsp47 did not substantially induce the activation of the UPR in HEK293T cells. Moreover, we used primary cortical neurons to determine the effect of depleting Hsp47 on UPR activation and endogenous GABAA receptors. Clearly, knocking down Hsp47 reduced the protein levels of endogenous GABAA receptors, including the major α1, β2/β3, and γ2 subunits, without activating the UPR in cortical neurons (Figure 6C). Therefore, since genetic manipulations of Hsp47 did not induce the activation of the UPR, the positive effect of Hsp47 on enhancing the GABAA receptor assembly and trafficking was not likely through the alteration of the global ER proteostasis network.”

2. Along the same lines, do stress-induced increases in HSP47 influence the assembly, trafficking, and activity of GABAA receptors in the context of a UPR? In other words, does increase HSP47 contribute to improved ER proteostasis of GABAA receptors observed upon the activation of UPR signaling pathways?

We determined the effect of activating ATF6 or IRE1/XBP1s on Hsp47 protein levels. Interestingly, in HEK293T cells expressing α1(A322D)β2γ2 receptors, activating ATF6 or XBP1s did not change Hsp47 protein levels, but increased α1(A322D) protein levels significantly (Figure 6D). Therefore, it appears that the effect of UPR activation on enhancing GABAA proteostasis was not through Hsp47 upregulation.

On page 18, we stated, “Furthermore, we determined the effect of activating the UPR on the expression of Hsp47 in HEK239T cells expressing misfolding-prone GABAA receptors carrying the α1(A322D) variant. Activating the ATF6 arm genetically by overexpressing the active ATF6-N fragment or pharmacologically by the application of two ATF6 activators, namely AA147 and AA263 [68], did not significantly change Hsp47 protein levels (Figure 6D), whereas as expected, such operations increased α1(A322D) and BiP protein levels (Figure 6D) [15,69]. In addition, activating the IRE1 arm genetically by overexpressing XBP1s did not increase Hsp47 protein levels (Figure 6D). These results indicated that activating the UPR did not change Hsp47 expression in HEK293T cells expressing a misfolding-prone GABAA receptor variant. Therefore, the effect of UPR activation on enhancing GABAA receptor proteostasis was not through Hsp47 upregulation.”

3. It would also be useful to show that ER proteostasis (i.e., folding, trafficking) of another protein is unaffected by HSP47 overexpression. This doesn't necessarily need to be a membrane protein but could give support to the idea that the observed effects are selective for Cys loop membrane proteins.

We tested the influence of Hsp47 on other ion channels that have different structural scaffolds compared to Cys-loop receptors, including NMDA (N-methyl-D-aspartate) receptors and hERG (human ether-a-go-go-related) potassium channels. In addition, we determined the effect of Hsp47 on an ERAD substrate using L444P β-glucocerebrosidase, the most common disease-causing variant in neuropathic Gaucher diseases, as a example. Our results indicated that Hsp47 did not positively regulate the protein levels of NMDA receptors (Figure 7—figure supplement 2A), hERG channels (Figure 7—figure supplement 2B), or L444P-β-glucocerebrosidase (Figure 7—figure supplement 2C).

Moreover, we tested the effect of Hsp47 on the biogenesis of another Cys-loop receptor family member, 5-HT_3_ receptors, which are assembled from 5-HT_3_A and 5-HT_3_B subunits. Indeed, our results demonstrated that Hsp47 positively regulates the surface expression of both 5-HT_3_A and 5-HT_3_B subunits (Figure 7D). Therefore, it appears that Hsp47 has certain selectively for Cys-loop receptors.

These new results were included on pages 19-20: “In addition, 5-HT_3_Rs are assembled from 5-HT_3_A and 5-HT_3_B subunits into heteropentamers in HEK293T cells [76]. Clearly, overexpressing Hsp47 increased the surface expression of both the 5-HT_3_A and 5-HT_3_B subunits according to surface biotinylation assay as well as their total protein levels in HEK293T cells (Figure 7D). These results indicated that Hsp47 has a general role in promoting the surface presence of the heteropentameric Cys-loop receptors.

Furthermore, we tested the influence of Hsp47 on other ion channels that have different structural scaffolds compared to Cys-loop receptors, including tetrameric NMDA (N-methyl-D-aspartate) receptors and hERG potassium channels. NMDA receptors, assembled from two obligatory GluN1 subunits and two GluN2 subunits, play a critical role in mediating synaptic development and plasticity in the central nervous system [77]. Overexpressing Hsp47 did not change or decreased the total or surface protein levels of WT GluN2A subunits (Figure 7—figure supplement 2A, lane 2 to lane 1) and those of misfolding-prone M705V GluN2A subunits [78] (Figure 7—figure supplement 2A, lane 4 to lane 3) in HEK293T cells, indicating that Hsp47 did not positively regulate NMDA receptor proteostasis. Moreover, hERG potassium channels regulate cardiac action potential repolarization in the heart, and loss of their function leads to type 2 long QT syndrome [49]. Knocking down Hsp47 did not change the protein levels of the mature (Figure 7—figure supplement 2B, top 155 kDa bands) and immature (Figure 7—figure supplement 2B, bottom 135 kDa bands) protein levels of WT hERG as well as trafficking-deficient hERG variants (N470D and T65P) in HEK293T cells, indicating that Hsp47 did not influence biogenesis of hERG channels. Furthermore, we determined the effect of Hsp47 on an ERAD substrate, L444P β-glucocerebrosidase, the most common disease-causing variant in neuropathic Gaucher diseases [79]. Knocking down Hsp47 did not change the total protein levels of folding-deficient L444P β-glucocerebrosidase in Gaucher patient-derived fibroblasts (Figure 7—figure supplement 2C), indicating that Hsp47 did not influence the biogenesis of an ERAD substrate in the ER lumen.

Therefore, our results indicated that Hsp47 had a general role in increasing the surface expression of heteropentameric Cys-loop receptors, including GABAA receptors, nAChRs, and 5-HT_3_Rs. Furthermore, Hsp47 had certain selectivity for Cys-loop receptors since Hsp47 did not enhance the biogenesis of other structurally diverse ion channels, such as NMDA receptors and hERG channels, or an ERAD substrate in the ER lumen, L444P β-glucocerebrosidase.”

4. It wasn't clear to me how the BiP binding to the GABAA subunit described in Figure 1C was performed. Was this in the presence of nucleotide? How can one compare the relative binding affinities of BiP and HSP47 without considering the nucleotide-dependent binding affinity of BiP. I'd more be careful with that comparison in the text.

We agreed that it is not appropriate to compare the binding affinity of BiP to that of Hsp47 without considering the ATP/ADP-dependent BiP binding. Since this manuscript focuses on characterizing Hsp47 for its role in the biogenesis of Cys-loop receptors, we removed the BiP MST binding data for future, more systematic study. Instead, in addition to showing the binding affinity of Hsp47 to α1(ERD) domain of GABAA receptors, we added the binding affinity of Hsp47 to β2(ERD) domain of GABAA receptors in Figure 1C.

On page 7, we stated, “we determined the binding affinity between Hsp47 and α1(ERD) or β2(ERD). A MicroScale Thermophoresis (MST) assay reported strong interactions between Hsp47 and the ERD of GABAA receptor subunits: Kd (Hsp47-α1(ERD)) = 102 ± 10 nM; Kd (Hsp47-β2(ERD)) = 127 ± 15 nM (Figure 1C). Therefore, Hsp47 binds to GABAA receptor subunits with high affinity.”

5. The authors convincingly demonstrate the importance of HSP47 in promoting the assembly of the GABAA receptors, but it isn't clear if HSP47 specifically is important for the assembly of a single interface or if all receptor interfaces require HSP47 for assembly. In other words, if HSP47 is depleted/overexpressed are specific assembly intermediates identified? It isn't clear from Figure 4C, although higher levels of HSP47 do appear to increase a band in the alpha1 blot (not observed in the beta2 blot) of ~120 kDa that could reflect a specific interaction.

We thank the review for pointing it out. Currently, we cannot conclude which assembly intermediates or receptor interfaces that Hsp47 acts on, which merits further investigations.

We added a discussion about this on page 21, “It is unclear whether Hsp47 targets certain subunit-subunit interfaces, such as α1-β2, α1-γ2, or β2-γ2 interfaces. In addition, the assembly intermediates that Hsp47 promotes, such as dimers and trimers, require future investigation.”

6. With the mutant GABAA subunit, the authors show increased trafficking and activity in response to HSP47 overexpression. How does this improved trafficking/activity compare to the wild-type protein?

We tested the effect of Hsp47 overexpression on additional three pathogenic α1 variations (Figure 5E), including S76R, D219N, and G251D. Overexpressing Hsp47 (Figure 5—figure supplement 2) increased the surface expression (Figure 5F) and GABA-induced peak current amplitudes (Figure 5G) for these three α1 variants.

On page 16, we stated, “Moreover, we evaluated the effect of Hsp47 on additional trafficking-deficient α1 variants, including α1(S76R), α1(D219N), and α1(G251D) (Figure 5E) [38]. Previously, it was reported that these variations decreased the surface expression of the α1 subunits and reduced GABA-induced peak current amplitudes to 33.3% for α1(S76R), to 60.3% for α1(D219N), and to 49.2% for α1(G251D) compared to wild type receptors [65]. Overexpressing Hsp47 (Figure 5—figure supplement 2) significantly promoted the surface expression of these α1 variants in HEK293T cells (Figure 5F). Furthermore, Hsp47 overexpression increased the peak currents 1.59-fold in HEK293T cells expressing α1(S76R)β2γ2 receptors, 1.72-fold in HEK293T cells expressing α1(D219N)β2γ2 GABAA receptors, and 1.87-fold in HEK293T cells expressing α1(G251D)β2γ2 GABAA receptors (Figure 5G), which are comparable to the peak currents for wild type receptors, suggesting the clinical potential of this approach. Furthermore, the effect of Hsp47 overexpression on increasing GABA-induced peak current amplitudes is more dramatic for trafficking-deficient α1 variants than for WT GABAA receptors (Figure 3—figure supplement 1A, Figure 5D, 5G).”

Reviewer #2 (Recommendations for the authors):Wang and colleagues provide evidence that the ER-resident HSP47 chaperone promotes the folding of GABA receptor subunits and the assembly of GABA subunits into multimeric ion channels. They demonstrate HSP47 can rescue the folding and function of a missense mutant A332D epilepsy-associated GABA subunit. They also demonstrate similar enhanced folding/function for acetylcholine receptor assembly. Overall, the experimental data in mouse brain lysates, cultured rat neurons, and HEK293 cells are well-presented and provide insight into new ion channel clients of the HSP47 chaperone and suggest that HSP47 expression can target disease-associated ion channel mutations.1. Authors show in Figure S1 model that unassembled or misfolded GABA subunits are degraded. Is HSP47 knock-down leading to degradation of GABA subunits, either for wild-type or the A332D mutant subunit, or are the GABA subunits trapped in a BiP-bound state without HSP47? Conversely, is over-expression reducing ubiquitination and ERAD of the GABA subunits?

We tested the effect of Hsp47 on the ubiquitination, apparent degradation rate, and a1-BiP interactions for both WT and a1(A322D). We showed that overexpressing Hsp47 reduced the ubiquitinated WT a1 (Figure 3—figure supplement 1B) and a1(A322D) proteins (Figure 5B). Interestingly, the apparent degradation rates were not significantly influenced by overexpressing or knocking down Hsp47 in both cases (Figure 3—figure supplement 1C, 1D and Figure 5—figure supplement 1A, 1B). In addition, knocking Hsp47 had minimal influences on the α1-BiP interaction in both cases (Figure 3—figure supplement 1C and Figure 5—figure supplement 1C), suggesting the involvement of additional ER proteostasis network in handing misfolded α1 proteins.

On pages 10, regarding wild type α1, we stated, “In addition, cellular ubiquitination assay demonstrated that Hsp47 overexpression decreased ubiquitinated α1 protein level (Figure 3—figure supplement 1B), suggesting that Hsp47 reduced the population of misfolded α1 proteins. Cycloheximide-chase assay showed that overexpressing Hsp47 (Figure 3—figure supplement 1C) or knocking down Hsp47 (Figure 3—figure supplement 1D) did not change the apparent degradation rate of α1 proteins significantly. Moreover, co-immunoprecipitation assay showed that knocking down Hsp47 did not significantly influence the interactions between α1 and BiP, an Hsp70 family chaperone in the ER lumen (Figure 3—figure supplement 1E), suggesting the involvement of additional ER proteostasis network components in handling misfolded α1 proteins.”

On page 15, regarding α1(A322D), we stated, “Consistently, Hsp47 overexpression substantially reduced the heavily ubiquitinated α1(A322D) protein (Figure 5B), indicating that Hsp47 decreased the population of misfolded α1(A322D) protein. Cycloheximide-chase assay demonstrated that overexpressing Hsp47 (Figure 5—figure supplement 1A) or knocking down Hsp47 (Figure 5—figure supplement 1B) did not change the apparent degradation rate of α1(A322D) significantly. In addition, co-immunoprecipitation assay showed that knocking down Hsp47 did not increase the interactions between BiP and α1(A322D) (Figure 5—figure supplement 1C), suggesting the involvement of additional proteostasis network components in handling misfolded α1(A322D).”

2. Authors show that HSP47 enhances folding and assembly for wild-type and A332D mutant GABA subunits. Authors show that, in the case of A332D, HSP47 overexpression enhances ion channel function. What about wild-type GABA subunits? Are there functional consequences for HSP47 overexpression on wild-type channel ion conductance, since they show improved folding and assembly?

Indeed, patch-clamping experiments showed that Hsp47 overexpression also increased peak currents for WT GABAA receptors in HEK293T cells (Figure 3—figure supplement 1A). On page 10, we stated, “Consistently, Hsp47 overexpression increased the peak current 1.6-fold in HEK293T cells expressing α1β2γ2 receptors (Figure 3—figure supplement 1A).”

In addition, we tested the effect of Hsp47 overexpression on additional three pathogenic α1 variations (Figure 5E), including S76R, D219N, and G251D. Overexpressing Hsp47 (Figure 5—figure supplement 2) increased the surface expression (Figure 5F) and GABA-induced peak current amplitudes (Figure 5G) for these three α1 variants.

On page 16, we stated, “Moreover, we evaluated the effect of Hsp47 on additional trafficking-deficient α1 variants, including α1(S76R), α1(D219N), and α1(G251D) (Figure 5E) [38]. Previously, it was reported that these variations decreased the surface expression of the α1 subunits and reduced GABA-induced peak current amplitudes to 33.3% for α1(S76R), to 60.3% for α1(D219N), and to 49.2% for α1(G251D) compared to wild type receptors [65]. Overexpressing Hsp47 (Figure 5—figure supplement 2) significantly promoted the surface expression of these α1 variants in HEK293T cells (Figure 5F). Furthermore, Hsp47 overexpression increased the peak currents 1.59-fold in HEK293T cells expressing α1(S76R)β2γ2 receptors, 1.72-fold in HEK293T cells expressing α1(D219N)β2γ2 GABAA receptors, and 1.87-fold in HEK293T cells expressing α1(G251D)β2γ2 GABAA receptors (Figure 5G), which are comparable to the peak currents for wild type receptors, suggesting the clinical potential of this approach. Furthermore, the effect of Hsp47 overexpression on increasing GABA-induced peak current amplitudes is more dramatic for trafficking-deficient α1 variants than for WT GABAA receptors (Figure 3—figure supplement 1A, Figure 5D, 5G).”

3. Authors propose that HSP47 preferentially binds the correctly folded conformation of GABA subunits. But for the A332D mutation, how can it adopt a correctly folded conformation given the inherent amino acid substitution? Can authors provide more insight into the consequences and A332D on GABA, and how HSP47 is overcoming the mutation?

We added more discussions about how Hsp47 overexpression could promote the functional surface expression of GABAA receptor variants.

On page 24, we discussed, “Since we envisage that Hsp47 interacts with proteins in the folded states or late-folding states and stabilizes the assembly intermediates and assembled receptor complex, overexpressing Hsp47 can “pull” more mutant subunits into the folded/assembled receptors that can engage the trafficking machinery for transport to the plasma membrane. Consequently, the function of the mutant receptors would be restored after Hsp47 overexpression. The rescuing mechanism of Hsp47 is similar to that of pharmacological chaperones, which bind to folded/assembled mutant GABAA receptors to stabilize their conformation to enhance their surface transport [65,90]. Indeed, we showed that Hsp47 overexpression as well as application of pharmacological chaperones, such as Hispidulin and TP003, enhanced the functional surface expression of a variety of trafficking-deficient α1 variants (Figure 5D, 5G) [65]. Therefore, this strategy serves as a proof-of-principle case for promoting the multi-subunit assembly process to ameliorate diseases resulting from membrane protein folding/assembly deficiencies.”

4. The functional studies of HSP47 overexpression are performed in HEK293 cells. Is it feasible to evaluate HSP47 overexpression in the native rat neurons on GABA ion channel conductance?

We carried out experiments in neurons and showed that overexpressing Hsp47 enhanced the surface staining of endogenous GABAA receptors (Figure 2B) and increased GABA-induced peak currents in rat hippocampal neurons (Figure 2C).

On pages 8-9, we stated, “overexpressing Hsp47 by transduction of lentivirus carrying *SERPINH1* cDNA significantly enhanced the surface staining of endogenous α1, β2/β3, and γ2 subunits in neurons (Figure 2B, row 2 to row 1).” “overexpressing Hsp47 increased the peak current amplitude to 2455 ± 406 pA in hippocampal neurons (Figure 2C). Collectively, the experiments in Figure 2 unambiguously reveal a novel role of Hsp47 as a positive regulator of the functional surface expression of endogenous GABAA receptors, an important neuroreceptor.”

Reviewer #3 (Recommendations for the authors):– In Figure 1B, it would be good to include a control of Hsp47 pull-down with another membrane protein to indicate specificity.

We included another two membrane proteins, hERG potassium channels and ZIP7 (an ER membrane zinc transporter), as negative controls to show that indeed Hsp47 did not interact with them (Figure 1—figure supplement 2A).

On page 7, we stated, “Moreover, recombinant Hsp47 did not interact with recombinant hERG (human ether-a-go-go-related) potassium channels [49], or recombinant ZIP7 (gene: *SLC39A7*), an ER membrane zinc efflux transporter [50] (Figure 1—figure supplement 2A), indicating that Hsp47 has certain selectivity to bind membrane proteins in vitro.”

– Overall, very little alpha1 appears to be EndoH resistant under EV conditions in HEK293 cells. Does this low trafficking efficiency track with rat primary neurons or brain tissue more broadly?

In HEK293T cells, 39% of WT α1 protein is Endo H resistant (Figure 3A, lane 2), indicating that a substantial fraction of WT α1 can exit the ER for anterograde trafficking (also see the next question).

– Can the authors explain what the middle band is between EndoH resistant and sensitive (Figure 3A, 5A)?

We added an explanation on page 9, “Since the α1 subunit has two N-glycosylation sites at Asn38 and Asn138, Endo H digestion generated two Endo H-resistant α1 bands (Figure 3A, lanes 2 and 4, top two bands), corresponding to singly and doubly glycosylated α1 proteins, which were observed in previous experiments [55].”

– For Figure 4B and 5A, a blot should be added to quantify Hsp47 overexpression.

Hsp47 blots were added to Figure 4B and Figure 5A, and Hsp47 overexpression was quantified and included in Figure 4B and Figure 5A.

– Please provide additional details for the FRET measurements of the homopentameric alpha7 nAChR complex assembly. Since alpha7 subunits are tagged with a mixture of fluorophores this would produce heterogenous complex mixtures. This should lower the sensitivity of the assay to detect changes in the assembly efficiency. How does the LOD change relative to heteromeric complexes?

We added an experimental detail for the FRET measurements of homopentameric α7 nAChR in the figure legend of Figure 7—figure supplement 1A.

In addition, we determined the effect of Hsp47 overexpression on the total protein levels of heteropentameric α4β2 nAChRs (Figure 7A) and homopentameric α7 nAChRs (Figure 7—figure supplement 1B) using Western blot analysis. On pages 18-19, we reported the different effects, “Heteropentameric α4β2 nAChRs and homopentameric α7 nAChRs are the major subtypes in the CNS [73]. Overexpressing Hsp47 significantly increased the total protein levels of α4 and β2 subunits in HEK293T cells (Figure 7A).” However, “In addition, Hsp47 overexpression did not change α7 total protein levels (Figure 7—figure supplement 1B). These results suggested that the capability of Hsp47 in the regulation of the biogenesis of α4β2 and α7 nAChRs is different.”